# Diketopyrrolopyrrole-based two-dimensional poly(arylene vinylene)s with high charge carrier mobility

Ruyan Zhao [1,2,9] ✉, Hongde Yu [2,9], Heng Zhang [3,9], Lei Gao [3,9], Arafat Hossain Khan [2], Congxue Liu [4], Xiaodong Li [1,2], Xingyuan Chu [2], Yubin Fu [1,2], Darius Pohl [5], Angelika Wrzesińska-Lashkova [6,7], Eike Brunner [2], Yana Vaynzof [6,7], Hai I. Wang [3,8], Mischa Bonn [3] ✉, Thomas Heine [2], Mingchao Wang [4] ✉ & Xinliang Feng [1,2] ✉

Layered two-dimensional conjugated polymers (2D CPs), or 2D conjugated covalent organic frameworks, are promising semiconductor materials for (opto)electronics and photocatalysis, but their performance is often limited by insufficient in-plane conjugation and poor charge transport. Guided by density functional theory calculations, we report two donor-acceptor-type 2D poly(arylene vinylene)s constructed from thienyl-benzodithiophene and diketo-pyrrolopyrrole units. These materials are predicted to exhibit strongly dispersive energy bands with ultralow in-plane effective masses ($0.036 - 0.159$ $m_0$), enabling intrinsic charge mobilities approaching 2000 $cm^2 V^{-1} s^{-1}$. Solid-state Aldol-type 2D polycondensation yields crystalline materials with optical band gaps as narrow as 1.0 eV. Terahertz spectroscopy reveals long charge carrier scattering times of 76 fs and a high room-temperature mobility of 310 $cm^2 V^{-1} s^{-1}$, surpassing previously reported linear and 2D CP powder samples. This work highlights donor-acceptor engineering as an effective strategy to enhance charge transport in 2D CPs.

Layered two-dimensional conjugated polymers (2D CPs), which are exemplified by 2D conjugated covalent organic frameworks (2D c-COFs) with extended in-plane π-conjugation and out-of-plane electronic couplings, are promising organic 2D crystal materials for applications in (opto)electronics, photocatalysis, and electrochemistry[1–5]. Representative examples include imine[6]-/pyrazine[7,8]-/vinylene[9,10]-linked or poly(benzimidazobenzophenanthroline) (BBL)-type[11] 2D c-COFs, commonly referred to as 2D polyimines,

2D polypyrazines, 2D poly(arylene vinylene)s (2D PAVs), and 2D BBLs, respectively[12]. Among these, 2D PAVs synthesized via Knoevenagel[9], Aldol-type (extended Knoevenagel)[13], Horner–Wadsworth–Emmons[14], or Wittig[15] polycondensation reaction have demonstrated superior chemical/thermal stability, as well as enhanced π-conjugation compared to their imine-linked counterparts. However, most reported 2D PAVs still suffer from limited in-plane conjugation, as evidenced by weak electronic band dispersion, leading to inefficient hopping

¹Max Planck Institute of Microstructure Physics, Halle, Germany. ²Faculty of Chemistry and Food Chemistry & Center for Advanced Electronics Dresden (cfaed), Technische Universität Dresden, Dresden, Germany. ³Max Planck Institute for Polymer Research, Mainz, Germany. ⁴State Key Laboratory of Advanced Waterproof Materials & Guangdong Provincial Key Laboratory of Nano-Micro Materials Research, School of Advanced Materials, Peking University, Shenzhen Graduate School, Shenzhen, China. ⁵Dresden Center for Nanoanalysis (DCN), Technische Universität Dresden, Dresden, Germany. ⁶Chair for Emerging Electronic Technologies, Technische Universität Dresden, Dresden, Germany. ⁷Leibniz Institute for Solid State and Materials Research Dresden, Dresden, Germany. ⁸Nanophotonics, Debye Institute for Nanomaterials Science, Utrecht University, Utrecht, The Netherlands. ⁹These authors contributed equally: Ruyan Zhao, Hongde Yu, Heng Zhang, Lei Gao. ✉e-mail: ruyazhao@mpi-halle.mpg.de; bonn@mpip-mainz.mpg.de; mingchao.wang@pku.edu.cn; xinliang.feng@mpi-halle.mpg.de

transport with charge carrier mobilities typically below $10\, cm^2V^{-1}s^{-1}$. To address this challenge, significant efforts have been devoted to engineering the 2D PAV backbone with electron-rich and π-extended building blocks such as benzotrithiophene[16,17], thienyl-benzodithiophene (TBDT)[18], pyrene[19,20], etc., aiming to strengthen the in-plane π-conjugation, narrow the band gaps, and enhance the charge transport. While such electron-rich 2D PAVs have achieved band-like transport, they typically still exhibit relatively large optical band gaps above 1.6 eV and moderate charge carrier mobilities in the powder form, often below $70\, cm^2V^{-1}s^{-1}$.

In contrast, donor-acceptor-type 2D PAVs offer distinctive electronic structures, characterized by tunable optical band gaps spanning from the visible to the infrared range. These materials exploit electron-donating and -withdrawing interactions between donor and acceptor units, which can establish independent electron and hole transport pathways, thereby helping to bypass charge transport barriers, mitigate defects, and reduce energetic disorder. This ultimately yields narrow-bandgap semiconductors with high charge carrier mobility. In most 2D conjugated polymers, the acceptor components—such as 4,7-diphenylbenzothiadiazole, phenyl/thiophene-flanked diketopyrrolo-pyrrole—often include rotatable phenyl rings, which compromise the planarity and rigidity of the polymer backbone, thereby impeding both intra- and interlayer charge transport[21–23]. Very recently, the incorporation of purely planar diketopyrrolopyrrole (DPP) unit, serving as a strongly electron-withdrawing building block, into honeycomb 2D PAVs has markedly enhanced π-conjugation, resulting in narrow optical band gaps[24,25]. We envision that extending this design strategy to DPP-based donor-acceptor 2D PAVs in a tetragonal lattice could overcome the cross-conjugation limitations of honeycomb architectures, leading to exceptional charge carrier mobility—an area that remains largely unexplored.

In this work, we report the DPP as a versatile acceptor incorporated into a thiophene-based 2D PAV backbone. Density functional theory (DFT) calculations identify the pivotal role of vinylene linkage in reinforcing the p-orbital interaction across the TBDT/DPP-based polymer backbone. The predicted electronic band structures for 2D PAVs reveal strongly dispersive energy bands with in-plane effective masses as low as $0.036–0.159\, m_O$, -50 times lower than the out-of-plane direction, suggesting predominant intralayer charge transport. The predicted intrinsic charge mobility for these materials reaches as high as $1800\, cm^2V^{-1}s^{-1}$. We thus synthesized two highly crystalline DPP-based donor-acceptor 2D PAVs, namely 2DPAV-TBDT-DPP-1 and 2DPAV-TBDT-DPP-2, via a solid-state Aldol-type polycondensation, which incorporate TBDT as the donor unit and DPP units with N-methyl (DPP-1) or N-hexyl (DPP-2) substitutions as the acceptors. These materials exhibit optical band gaps as narrow as 1.0 eV, the smallest among reported 2D CPs. Optical pump-THz probe (OPTP) spectroscopy reveals charge carrier scattering times of 76 fs at room temperature, yielding a charge mobility of $310\, cm^2V^{-1}s^{-1}$, superior to the reported linear CPs, 2D CPs, COFs, and metal-organic frameworks (MOFs) tested by terahertz (THz) spectroscopy in the powder form. This work highlights the crucial role of rational donor-acceptor design in developing high-mobility and narrow-bandgap 2D CPs, and showcases their potential in future electronic applications.

## Results

### Design of DPP-based donor-acceptor polymer backbone

We selected TBDT and DPP as the building blocks to construct donor-acceptor 2D CPs[23,25–27]. DFT calculations were performed on a series of TBDT/DPP-based model compounds (MX, X = 1 – 7; Fig. 1a, b) bridged by aromatic rings commonly used in polyarylenes (including phenyl, pyridine, pyrazine, furan, thiophene, selenophene) or vinylene moieties in PAVs (see the details in Supplementary Figs. 1–4). Substituting the phenyl ring (M1) with pyridine (M2) and with pyrazine (M3), the planarity of model compounds gradually increased due to reduced

hydrogen repulsion (Supplementary Fig. 1). The electron-withdrawing nature of pyridine and pyrazine leads to downshifted LUMO energy levels in M2 and M3, and thus a reduction of HOMO-LUMO (HOMO = the highest occupied molecular orbital, LUMO = the lowest unoccupied molecular orbital) energy gap by 0.14 and 0.28 eV, respectively, compared to 2.36 eV for M1 (Fig. 1c). The incorporation of electron-rich furan, thiophene, or selenophene rings enhances the planarity, which results in upshifted HOMO energy levels and thus decreased HOMO-LUMO gaps by 0.33–0.43 eV in M4, M5, and M6, compared to M1. It is notable that linking the TBDT and DPP units by a vinylene bridge in M7 leads to the formation of vinylene H···O hydrogen bonds. This strengthens the planarity and p-orbital interaction, resulting in the smallest HOMO-LUMO gap (1.91 eV) among the calculated model compounds (Fig. 1d and Supplementary Fig. 5; see more details in Supplementary Figs. 6–10)[25]. Based on these findings and in view of synthetic feasibility, we propose that developing TBDT/DPP-based (M7 type) 2D PAVs via Aldol-type polycondensation would confer efficient 2D conjugation and enhanced charge transport to the 2D polymer frameworks.

### Predicted band structure and charge carrier mobility of DPP-based donor-acceptor 2D PAVs

We investigated the intrinsic charge transport properties of TBDT/DPP-based 2D PAVs by calculating their electronic band structures and projected density of states (PDOS) using the Perdew–Burke–Ernzerhof (PBE) functional[28] within the DFT framework. A N-methyl-substituted DPP (DPP-1) and a methyl-substituted TBDT were selected to simplify the model for computation (see chemical structure in Supplementary Fig. 11; the resulting model polymer is referred to as 2DPAV-TBDT-DPP-0, Fig. 1a). The monolayer 2DPAV-TBDT-DPP-0 is predicted to be a direct band-gap semiconductor with strongly dispersive conduction band maximum (CBM) and valence band minimum (VBM) (Fig. 2a), indicative of highly delocalized charge carriers. The calculated band dispersion reaches 0.57 eV for the CBM and 0.22 eV for the VBM (Fig. 2a), suggesting dominant electron transport characteristics (i.e., n-type behavior), unprecedented among the reported organic 2D crystals[3]. This high dispersion leads to ultrasmall effective masses of $0.072\, m_O$ and $0.159\, m_O$ for electrons and holes, respectively, for monolayer 2DPAV-TBDT-DPP-0 (Supplementary Table 3). These values are smaller than those of the reported high-mobility 2D CPs (e.g., monolayer effective mass of ca. $0.3\, m_O$ for electrons or holes[11]). According to the elemental-resolved PDOS diagram in the right panel of Fig. 2a, b, the p orbitals of carbon and oxygen atoms make substantial contributions to both the VBM and CBM. This indicates that the electron-deficient DPP unit is significant in the PDOS, and the highly conjugated backbone enhances electronic communication between the donor and acceptor units.

Similarly, the layer-stacked 2DPAV-TBDT-DPP-0 exhibits strong in-plane band dispersion, significantly exceeding that in the out-of-plane direction. Consequently, the estimated effective masses of 0.062 and $0.078\, m_O$ for electrons and holes, along the 2D plane, are -50 times lower than those of 3.455 and $3.457\, m_O$ in the out-of-plane direction, indicating predominant in-plane charge transport[29,30] Further calculations of intrinsic charge carrier scattering times (details in Supplementary Information) enable estimation of charge mobilities as high as $1800\, cm^2V^{-1}s^{-1}$ for electrons and $110\, cm^2V^{-1}s^{-1}$ for holes in the layer-stacked structure (Fig. 2c). To our knowledge, such electron-dominated transport with predicted superior charge mobility has not been previously reported for 2D CPs, COFs, or metal-organic frameworks (Supplementary Table 4).

Additionally, we investigated the impact of side chains on the charge transport properties. As shown in Fig. 2d, e, the 2D PAV based on hexyl-substituted TBDT and DPP-1 (2DPAV-TBDT-DPP-1, see the chemical structure in Supplementary Fig. 11 and Fig. 3a) presents band structures comparable to those of 2DPAV-TBDT-DPP-0. In contrast, the

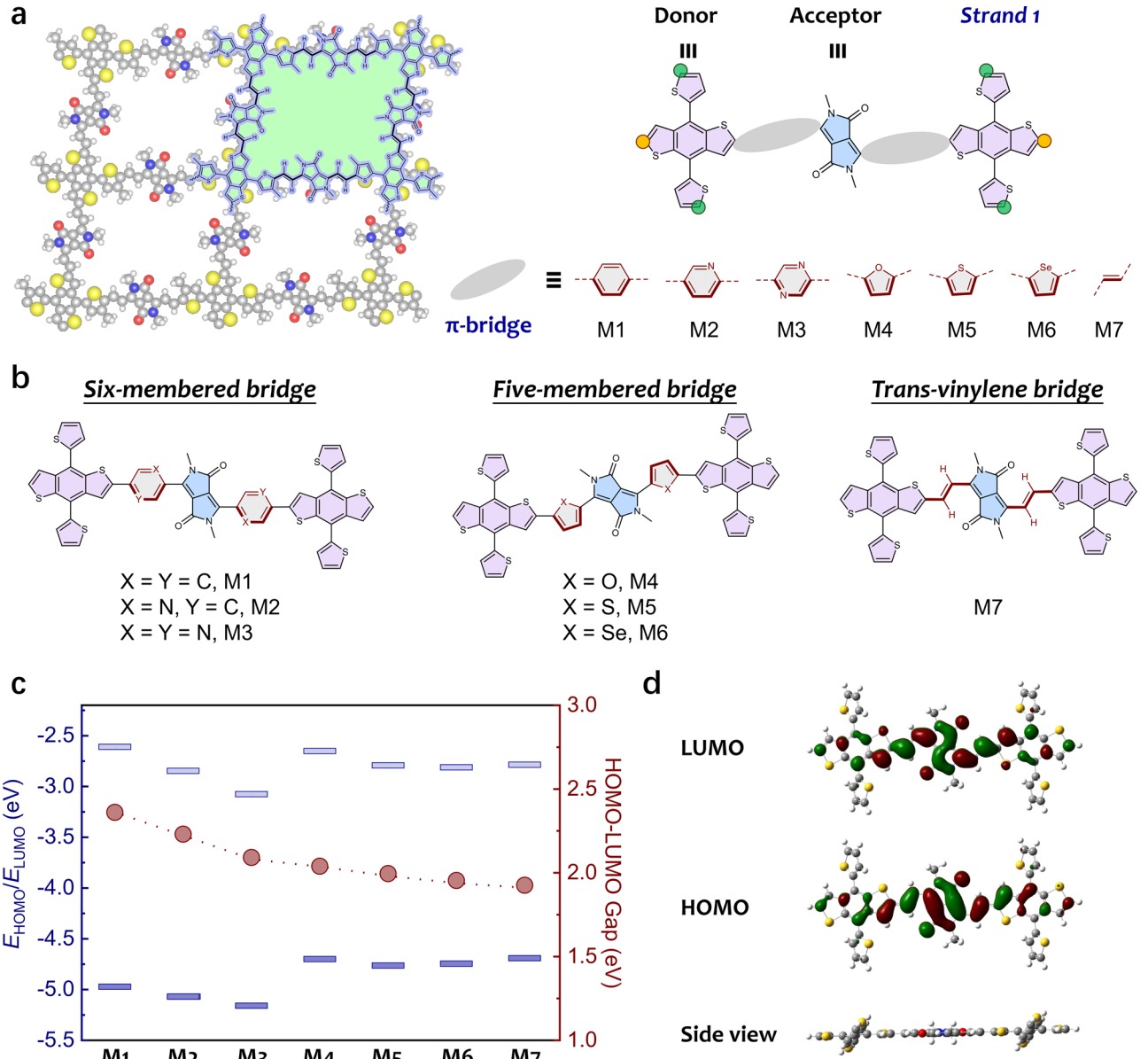

**Fig. 1 | Molecular design of DPP-based donor-acceptor 2D CPs. a** Representative structural models of TBDT/DPP-based 2D CPs and model compounds M1–M7. The two strands form along the benzodithiophene (yellow balls) and thienyl (green balls) sites, respectively. **b** Chemical structures of the model compounds M1–M7. **c** HOMO-LUMO energy levels and gaps of M1–M7. **d** Molecular frontier orbitals and side-view geometry of M7.

similar *N*-hexyl-DPP (DPP-2)-based 2DPAV-TBDT-DPP-2 shows slight backbone distortion due to steric hindrance from the multiple long alkyl chains. Taking monolayer 2D PAVs as examples, despite having similar in-plane CBM/VBM band dispersions (-0.58/0.25 eV for 2DPAV-TBDT-DPP-1 and 0.55/0.25 eV for 2DPAV-TBDT-DPP-2, see details in Fig. 2d and Supplementary Fig. 15a), 2DPAV-TBDT-DPP-2 exhibits an enlarged band gap. Moreover, layer-stacked 2DPAV-TBDT-DPP-2 shows negligible out-of-plane dispersion (Supplementary Fig. 15b), which is attributed to the enlarged interlayer spacing induced by the hexyl chains. The effective masses of electrons and holes exhibit a strong correlation with the average torsion angles of the polymer backbone along the *a* and *b* directions. The variation in their mass ratios closely follows the trend of the corresponding torsion angle ratios (Supplementary Figs. 16 and 17), suggesting that higher backbone planarity leads to lower effective mass, whereas deviations from planarity induce pronounced charge-transport anisotropy. The corresponding in-plane electron/hole effective masses for layer-stacked 2DPAV-TBDT-DPP-1 and 2DPAV-TBDT-DPP-2 are calculated to be

0.036/0.087 $m_0$ and 0.069/0.148 $m_0$, respectively (Supplementary Table 3).

## Synthesis and characterization of DPP-based donor-acceptor 2D PAVs

We then synthesized the 2D PAVs from corresponding monomers (TBDT-1 and DPP-1 or DPP-2) (Fig. 3a) using sodium benzoate as the catalyst in a solid-state manner at 200 °C for 3 days. The crystalline nature of 2D PAVs was resolved by powder X-ray diffraction (pXRD) analysis (Fig. 3b). Taking 2DPAV-TBDT-DPP-1 as an example, the pXRD pattern shows diffraction peaks at 5.10°, 6.60°, 10.33°, 15.60°, and 25.76° that are ascribed to (100), (110), (200), (300), and (001) facets, respectively. The full width at half maximum of the (100) facet diffraction is as small as 0.39°, comparable to those of highly crystalline imine-linked 2D c-COFs, suggesting its high crystallinity (Supplementary Table 6)[31]. The interlayer distance derived from the (001) peak is 3.46 Å. Structure modeling was carried out to elucidate the packing structure of the 2D PAVs. The calculated pXRD pattern of a slipped-AA

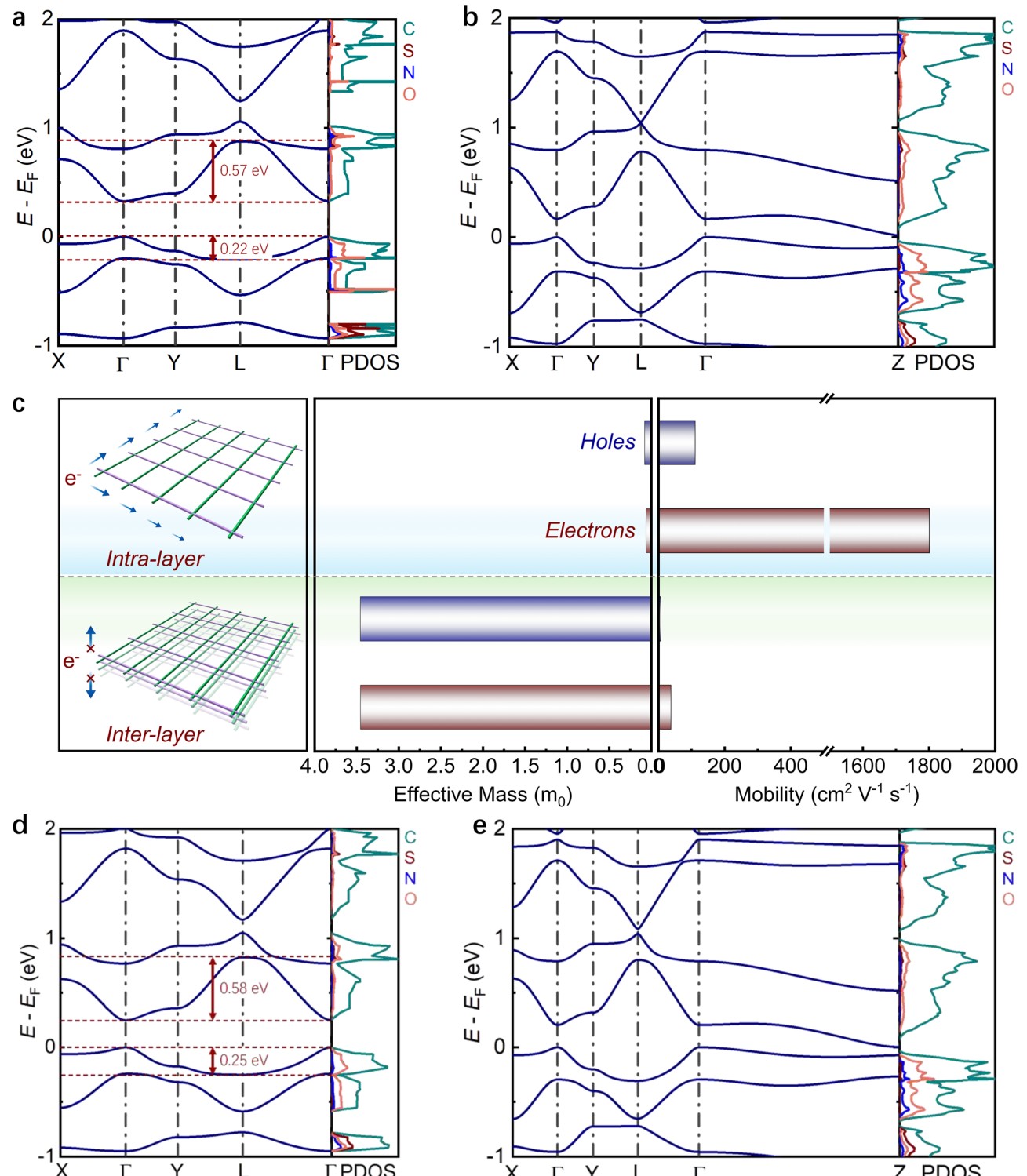

**Fig. 2 | Predicted band structure and charge mobility of DPP-based 2D PAVs. a**, **b** Electronic band structures and PDOS (C*p*, N*p*, O*p*, and S*p*) of monolayer and layer-stacked 2DPAV-TBDT-DPP-0, respectively. **c** Calculated effective masses and charge mobilities for electron (red) and hole (blue) in the in-plane or out-of-plane direction. **d**, **e** Electronic band structures and PDOS (C*p*, N*p*, O*p*, and S*p*) of monolayer and layer-stacked 2DPAV-TBDT-DPP-1, respectively.

model (Fig. 3c, d) matches well with the experimental signals. Pawley refinement shows a small $R_{wp}$ value of 2.38% and $R_p$ value of 1.30% (Supplementary Fig. 19). The lattice parameters are obtained as $a = 17.69$ Å, $b = 21.81$ Å, $c = 3.89$ Å, $\alpha = 82.68°$, $\beta = 80.84°$, $\gamma = 91.33°$.

2DPAV-TBDT-DPP-2 was synthesized following the same procedure. It displays a similar pXRD pattern to 2DPAV-TBDT-DPP-1, but with a shift of the predominant diffraction from the (100) to the (010) plane

at a lower angle of 4.31°, corresponding to a larger *b*-lattice parameter than *a*. 2DPAV-TBDT-DPP-2, with its longer hexyl (C6) chains compared to the methyl (C1) chains in 2DPAV-TBDT-DPP-1, exhibits a slight lattice expansion, particularly along the *c*-direction (+0.3 Å), with smaller changes along the *a*-direction (+0.1 Å) and *b*-direction (−0.3 Å). These changes result in little shifts in diffraction angles for the (100) and (010) peaks. We note that interlayer sliding in these COFs can

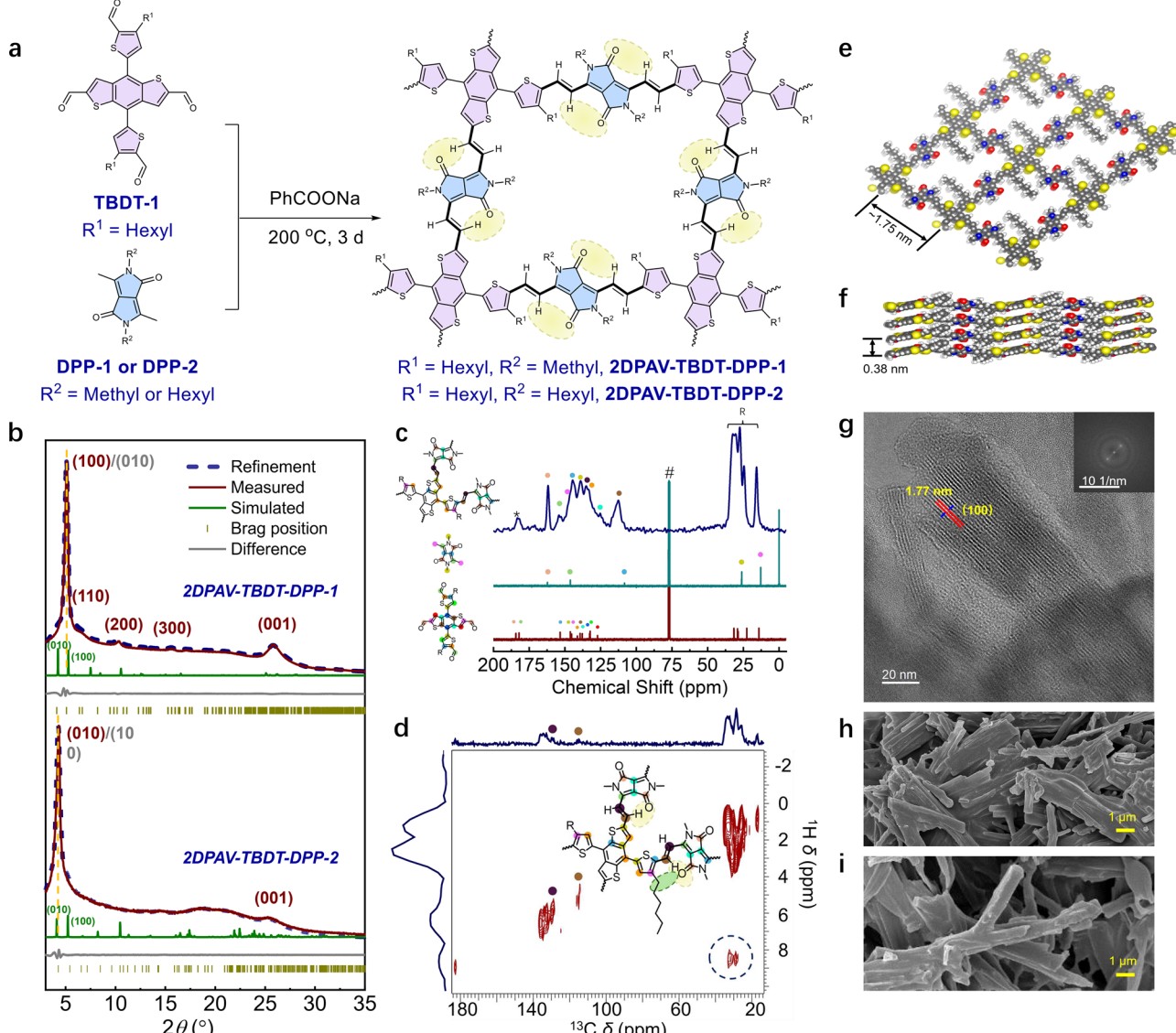

**Fig. 3 | Synthesis and characterization of TBDT/DPP-based crystalline 2D PAVs. a** Schematic synthesis. The yellow circles indicate the potentially hydrogen bondings. **b** Experimental, Pawley-refined and simulated pXRD patterns as well as the difference plots of 2DPAV-TBDT-DPP-1 (top) and 2DPAV-TBDT-DPP-2 (bottom), respectively. **c** Solid-state [13]C CP MAS NMR spectra of 2DPAV-TBDT-DPP-1. The liquid-state NMR spectra of the monomers are shown for comparison. **d** [1]H–[13]C HETCOR spectra of 2DPAV-TBDT-DPP-1. **e, f** Top and side views of the structure model. **g** HR-TEM image and the inset for the SAED pattern. SEM images of 2DPAV-TBDT-DPP-1 (**h**) and 2DPAV-TBDT-DPP-2 (**i**).

further influence the positions of these peaks, contributing to the observed diffraction patterns. (Fig. 3b; see the Pawley refinement results in Supplementary Fig. 20). Besides, the diffraction peak attributed to π-π stacking shifts from 25.76° to 25.30° in 2DPAV-TBDT-DPP-2. This corresponds to a slightly increased interlayer distance of 3.52 Å, suggesting weakened π-π interactions. Additionally, we performed high-resolution transmission electron microscopy (HR-TEM) and selected-area electron diffraction (SAED) measurements to reveal the crystalline nature of the 2D PAV. For example, a lattice fringe with a distance of ~1.77 nm is observed for 2DPAV-TBDT-DPP-1 (Fig. 3g), which corresponds to its (100) facet and agrees well with the above results.

Solid-state cross-polarization magic angle spinning nuclear magnetic resonance (CP MAS NMR) and Fourier-transform infrared (FT-IR) spectroscopies were utilized to identify the chemical structure of the 2D PAVs. Depth [1]H-NMR spectrum displays broad proton signals at 6–8 and 0–4 ppm, attributed to aromatic and aliphatic protons, respectively (Supplementary Figs. 21 and 22)[32]. [13]C CP MAS NMR spectra

display distinct carbon signals for the trans-vinylene (135 and 113 ppm), TBDT (147, 144, 139, and 132 ppm), DPP (162 ppm for imide carbon) moieties, as well as signals in the range of 0–50 ppm for the aliphatic carbons (see the detailed peak assignment in Fig. 3e and Supplementary Fig. 23). A small signal at 180 ppm can be attributed to the unreacted -CHO groups that may remain at the edge of the 2DPAV-TBDT-DPP-1 and 2DPAV-TBDT-DPP-2 particles (Supplementary Figs. 21 and 22, the ratios of aldehyde protons are determined to be 1.57% and 1.01%, respectively).

The formation of hydrogen bonds between vinylene hydrogen and DPP oxygen was disclosed in [1]H–[13]C heteronuclear correlation (HETCOR) spectroscopy. The proton signal corresponding to the vinylene linkage appears at 9 ppm due to hydrogen bonding and shows correlation with the spatially closest carbon (marked by a green circle) of the flexible aliphatic chain (blue dashed line circled area in Fig. 3d). As shown in the FT-IR spectra (Supplementary Figs. 24, 25 and 26), the characteristic C–H stretching vibration band at ~2987 cm⁻¹ from the active -CH₃ and the aldehyde C=O stretching

vibration band at ~1659 cm⁻¹ from the DPP and TBDT monomers, respectively, disappear; while new bands emerge at ca. 1562 cm⁻¹ and 960 cm⁻¹ attributable to the trans-vinylene bond stretching vibration. These results indicate the successful formation of the vinylene linkages in both 2D PAVs.

Scanning electron microscopy (SEM) images suggest that the as-synthesized 2D PAV powder samples exhibit a rod-like morphology with average lengths of several micrometers (Fig. 3h, i, and Supplementary Figs. 27 and 28). The 2DPAV-TBDT-DPP-2 rods are more extended in length than the 2DPAV-TBDT-DPP-1 rods, revealing its higher crystallinity or polymerization degree. This result can be attributed to the lower melting point and, thus, better flexibility of the DPP-2 monomer, which facilitates the solvent-free polycondensation reaction[33]. Energy-dispersive X-ray spectroscopy (EDS) mapping images confirm that the C, N, O, and S elements are uniformly distributed throughout the polymer nanocrystals (Supplementary Fig. 23). Moreover, the synthesized 2D PAVs show excellent chemical and thermal stabilities (see details in Supplementary Figs. 29–31).

## Optical and electronic properties

The optical properties of 2D PAVs powder were investigated by diffuse reflectance spectra using UV-vis-near IR absorption spectroscopy. Benefiting from the strong electron-pushing and -pulling effect between the TBDT and DPP units, both 2D PAVs exhibit broad optical absorption across the whole visible spectrum with tails extended to the near-IR region (Supplementary Table 7). The absorption maximum of 2DPAV-TBDT-DPP-1 is located at ~910 nm (Fig. 4a). The optical band gap is determined to be ~1.0 eV by means of Tauc plots (Fig. 4b), a value smaller than the reported 2D PAVs (wider than 1.6 eV) and 2D c-COFs[34]. By contrast, 2DPAV-TBDT-DPP-2 shows a larger optical band gap of 1.2 eV (Fig. 4b), which we tentatively ascribe to its enlarged interlayer distance and thus less pronounced interlayer coupling[3,35].

We further conducted ultraviolet photoemission spectroscopy (UPS) on 2DPAV-TBDT-DPP-1 film grown on an ITO substrate (Fig. 4c; see details about sample preparation in Supplementary Information). The work function (WF), calculated from the secondary electron cutoff ($E_{SECO}$ = 16.8 eV), was determined to be ~4.4 eV. The inset highlights the HOMO onset, extracted as 0.8 eV, resulting in an ionization potential (IP) of 5.2 eV. Using the ionization potential (IP) and the optical band gap (not accounting for the exciton binding energy), the electron affinity (EA) is estimated to be 4.2 eV. This energetic structure shows that the Fermi level is positioned close to the LUMO (Fig. 4d), indicating an n-type semiconductor nature. The low IP and small band gap indicate effective delocalization of π-electrons throughout the conjugated polymer backbone.

## THz photoconductivity measurements

Next, we investigated the charge transport properties of 2DPAVs by non-invasive ultrafast OPTP spectroscopy. OPTP offers a contact-free method for measuring the high-frequency, time-resolved conductivity of materials following the optical injection of charge carriers. Figure 5a shows the excitation fluence-dependent carrier dynamics of 2DPAV-TBDT-DPP-1 powder at room temperature over a 15-ps window. The peak of the photoconductivity increases linearly with excitation fluence (Fig. 5b), excluding strong carrier-carrier interactions and recombination processes. The dynamics within the first few ps is dominated by the free carrier response, with the photoconductivity decay caused by exciton formation or charge trapping[11,36].

The frequency-resolved photoconductivity spectra shown in Fig. 5c, d, measured 1.5 ps after the excitation pulse, confirm the free carrier response. The experimental data are well described by the DS model[8], from which we extract the scattering time ($\tau_s$) and back-scattering probability (c parameter). Charge mobility can be thus calculated by $\mu = e\tau_s(1+c)/m^*$ in the dc limit. The fitting for 2DPAV-TBDT-DPP-1 yields a $\tau_s$ of 36 ± 5 fs and a c parameter of −0.93 ± 0.02.

Considering the calculated electron-hole reduced effective mass (Supplementary Table 4), the charge mobility is estimated to be 170 ± 40 cm² V⁻¹ s⁻¹ at room temperature. In comparison, the fitting for 2DPAV-TBDT-DPP-2 gives a longer $\tau_s$ of 76 ± 4 fs and a larger c of −0.89 ± 0.02 (lower backscattering), which we attribute to its superior crystallinity associated with the higher polymerization degree characterized above, that is, reduced grain boundary and defect densities[37,38]. Taken together the electron-hole reduced effective mass, 2DPAV-TBDT-DPP-2 presents charge mobility as high as 310 ± 50 cm² V⁻¹ s⁻¹ at room temperature. Such a room-temperature mobility value represents a record value for the thus-far reported linear CPs, 2D CPs, COFs, and MOFs in powder form (Fig. 5e)[8,15,18,39,40]. Moreover, we studied the electrical transport properties of the 2D PAVs, which showed boosted electrical conductivities by two orders of magnitude via n-type doping (n-dopants: N-DMBI: [4-(1,3-dimethyl-2,3-dihydro-1H-benzoimidazol-2-yl)-phenyl]-dimethyl-amine; TDAE: tetrakis(dimethylamino)ethylene) compared to the pristine materials (see details in Supplementary Fig. 35 and Supplementary Table 9).

## Discussion

In conclusion, we have demonstrated two donor-acceptor type 2D PAVs with small band gaps and high charge carrier mobilities. By deliberately bridging the electron-donating building block of TBDT and the electron-deficient DPP unit via trans-vinylene bonds through a dynamic Aldol-type polycondensation methodology, we have achieved highly crystalline 2D PAVs with planar polymer backbone, thereby facilitating delocalization of π-electrons. The most dispersive electronic band structures, especially the long-unrealized dispersion in CBM, which is promising for efficient electron transport, are obtained. In addition, side-chain engineering on the DPP unit offers another leverage to fine-tune the crystallinity, morphology, and the interlayer electronic coupling of the final 2D PAVs. The broad optical absorption and state-of-the-art charge carrier mobilities, on the order of hundreds of cm² V⁻¹ s⁻¹ at room temperature, position these materials as highly promising for future electronic applications. Further structural design to develop highly crystalline or single-crystalline 2D PAVs[41] and improve their processability will not only enable a fundamental understanding of the structure-property relationships but also pave the way for their integration into organic (opto)electronic devices.

## Methods

### Prediction of electronic band structure and charge carrier mobility

The electronic properties of the 2D PAVs were calculated using density functional theory (DFT) with the Vienna Ab Initio Simulation Package (VASP 5.4.4)[42], implementing the projector-augmented wave (PAW)[43] method and the Perdew–Burke–Ernzerhof (PBE)[28] functional. To account for London dispersion forces, Grimme's D3 correction was applied[44]. For lattice parameter optimization, a plane-wave cutoff energy of 400 eV was used, while a higher cutoff energy of 600 eV was employed for static calculations. The force convergence criterion during optimization was set to 0.02 eV Å⁻¹, and the energy convergence criterion in the self-consistent field iterations was 10⁻⁵ eV for optimization and 10⁻⁶ eV for static calculations. Partial charge density plots were generated using VESTA[45]. A k-point mesh of 2 × 2 × 4 was utilized for structural optimization, 2 × 2 × 8 for converged charge density calculations, and 5 × 5 × 10 for density of states (DOS) calculations. The relaxation times of electrons and holes in 2DPAV-TBDT-DPP-0 are estimated to be 1.97 × 10⁻¹³ and 1.52 × 10⁻¹⁴ s.

### Synthesis of 2DPAV-TBDT-DPP-1 and 2DPAV-TBDT-DPP-2

A 5 mL high-pressure glass tube was charged with TBDT-1 (30.00 mg, 0.047 mmol), DPP-1 (18.17 mg, 0.095 mmol), and sodium benzoate (27.24 mg, 0.189 mmol). The tube was sonicated at room temperature

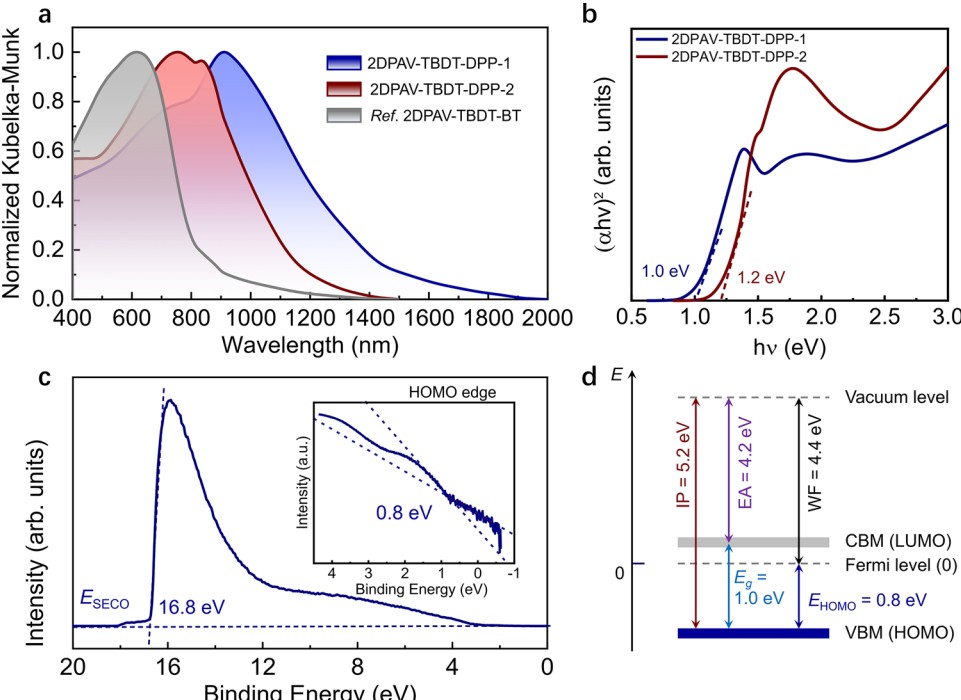

**Fig. 4 | Optical properties of 2D PAVs. a** Normalized diffuse reflectance spectra. A TBDT-based electron-rich 2DPAV-TBDT-BT is shown as a reference[32]. **b** Tauc plots. **c**, **d** UPS spectrum and energy level diagram of 2DPAV-TBDT-DPP-1, respectively.

$E_{SECO}$: the secondary electron cutoff; IP: the ionization potential; EA: the electron affinity; WF: the work function.

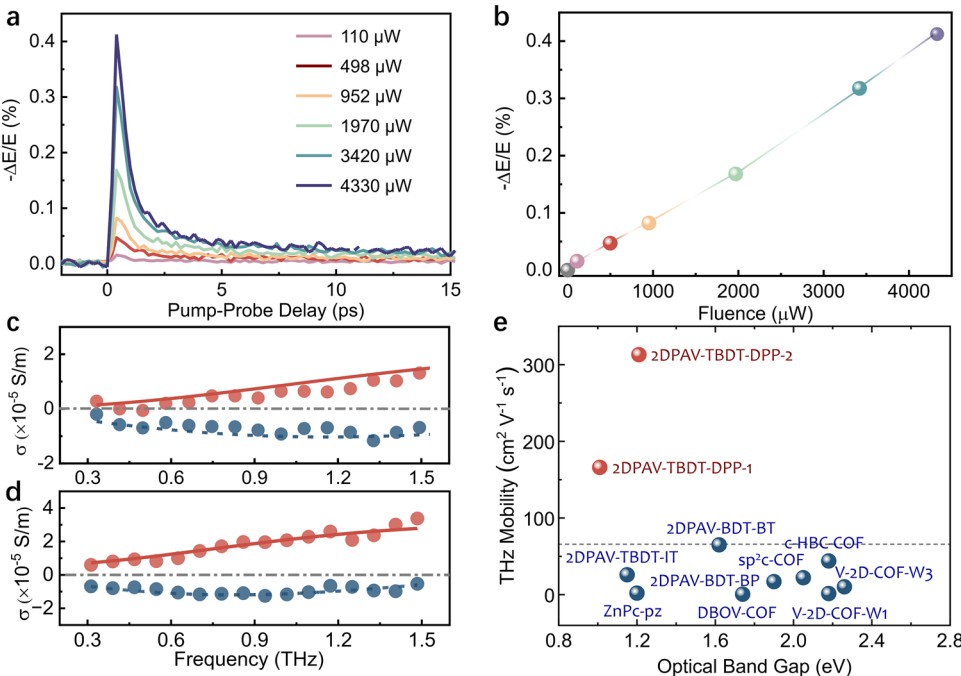

**Fig. 5 | Charge transport properties characterized by THz spectroscopy. a** Photoconductivity dynamics of 2DPAV-TBDT-DPP-1 at room temperature following excitations at 400 nm with different fluences. **b** The peak of photoconductivity in **a** as a function of fluence. **c**, **d** Frequency-resolved complex (red: real part; dark cyan: imaginary part) THz photoconductivity spectra of 2DPAV-

TBDT-DPP-1 and 2DPAV-TBDT-DPP-2, respectively, recorded 1.5 ps after excitation. The solid lines show the Drude-Smith fitting. **e** Comparison of optical band gaps and charge mobilities characterized by THz spectroscopy of reported 2D CPs and 2D c-COFs in the powder form.

for 3 min, degassed by three freeze-pump-thaw cycles, sealed under vacuum, and heated at 200 °C for 3 days. After cooling to room temperature, the precipitate was filtered, washed with dimethylformamide, acetone, water, tetrahydrofuran, and anhydrous acetone,

respectively, then collected and dried under vacuum at 100 °C overnight to get 2DPAV-TBDT-DPP-1 as a dark powder in 94% yield. Under the same conditions, TBDT-1 (30.00 mg, 0.047 mmol), DPP-2 (31.42 mg, 0.095 mmol), and sodium benzoate (27.24 mg, 0.189 mmol)

were used to synthesize 2DPAV-TBDT-DPP-2 as a dark powder in 96% yield.

## Optical pump-THz probe (OPTP) spectroscopy
In the OPTP measurement, the sample is first photoexcited by a 400 nm laser pulse (pulse duration 100 fs) to generate electrons and holes in the conduction and valence bands, respectively. Then, a single-cycle THz pulse with a pulse duration of 1 ps arrives, interacting with the photoexcited charge carriers, leaving a charge transport fingerprint in the transmitted THz pulse. The relative attenuation of the THz electric field $-\Delta E/E$ is proportional to the real part of the photoconductivity, while the phase shift is related to the imaginary part. The generation and recombination dynamics are obtained by changing the relative delay between the optical pulse and the THz pulse. The real part of photoconductivity increases with frequency, while the imaginary part decreases first and then increases at higher frequencies. This aligns with the response of free carriers experiencing spatial confinement due to grain boundaries or interfaces[46,47]. This type of charge transport can be described by the so-called Drude-Smith (DS) model:

$$\sigma(\omega) = \frac{\omega_p^2 \varepsilon_0 \tau_s}{1 - i\omega\tau_s}(1 + \frac{c}{1 - i\omega\tau_s})$$

Here, $\omega_p$ is the plasma frequency, $\varepsilon_0$ is the static dielectric constant, $\tau_s$ is the scattering time, $c$ ($-1 \leq c \leq 0$) describes the backscattering probability of charge carriers due to the confinement. If $c$ is 0, the DS model reverts to the Drude model with randomized momentum scattering; when $c$ approaches $-1$, the free charges are subject to strong preferential backscattering.

## Data availability
The data supporting the findings of this study are available within the paper and its Supplementary Information files. The data generated in this study are provided in the Supplementary Information/Source Data file. Source data are provided with this paper.

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

## Acknowledgements

This work was financially supported by National Natural Science Foundation of China (52572196), ERC Grant (T2DCP, No. 819698) and DFG projects (CRC 1415, No. 417590517; SPP 2248, RACOF-MMIS). We appreciate the Materials Processing and Analysis Center at Peking University, and the Analysis and Testing Center of School of Advanced Materials, Peking University Shenzhen Graduate School for assistance with materials characterization. We thank using facilities at the Dresden Center for Nanoanalysis (DCN). The authors acknowledge Mr. Wei Wang for help with conductivity measurement. The authors thank Mr. Tianhao Xue and Prof. Thomas Bein for the measurement of diffuse reflectance spectra. The authors also acknowledge the Centre for Information Services and High Performance Computing (ZIH) at TU Dresden for the provided computational resources.

## Author contributions

M.W., R.Z., and X.F. conceived and designed the project. R.Z. synthesized the monomers, prepared the 2D PAVs, and arranged the structural, compositional, and property characterizations. H.Y. and T.H. performed the theoretical calculations of the 2D PAVs. H.Z., L.G., H.-I.W., and M.B. conducted the THz experiments and data analysis. A.H.K. and E.B. conducted the solid-state NMR measurement. C.L., X.L., X.C., and D.P. contributed to the structural characterization of materials. Y.F. contributed to helpful discussions. A.W.-L. and Y.V. performed UPS. R.Z., M.W., and X.F. co-wrote the manuscript with contributions from all co-authors.

## Funding

## Competing interests

The authors declare no competing interests.
