## [Transparent Peer Review file · Nature Communications]

Diketopyrrolopyrrole-Based Two-Dimensional Poly(Arylene Vinylene)s with High Charge Carrier Mobility

Corresponding Author: Professor Xinliang Feng

Version 0:

Reviewer comments:

Reviewer #1

(Remarks to the Author)

X. Feng and co-workers report two new highly conjugated 2D COFs demonstrating an exceptionally small band-gap (1.0 eV) and high intrinsic electron mobility (as shown by DFT and THz measurements). The molecular design takes advantage of the 2D-conjugated building block TBDT-1 explored by the authors in their previous work (refs. 18, 32) and the recently reported aldol polymerization with dimethyl-DPP (Z. Zhang et al., ACIE 2025, e202417805; this paper should be mentioned in the Introduction). While the low band-gap of DPP-linked conjugated polymers is not itself surprising based on the 1D conjugated polymers literature, the very high charge mobility (especially for electrons) is nonetheless remarkable. I suspect that the high polymerization temperature (200°C) might limit applications of these COF (due to expected high defect density and difficulty of growing films), but the reported materials still stand out as some of the most promising non-fused semiconducting 2D polymers reported to date. I recommend accepting the paper after a revision of the following points.

- 1) The paper is a bit difficult to follow because the structures of the reported COFs as well as of the models M1-M7 are not shown in the manuscript and have to be guessed from the provided chemical names (or be dug from the SI). Overall, I feel that too much of essential information is delegated to the SI. On a minor point, I encourage the authors to move the figures from the end of the paper to within the text, to facilitate the reviewer's job.
- 2) The claim of the lowest E_g (among 2DCPs) should be supported by SI tables summarizing the current state-of-art on that matter. The SI table 4 comparing the calculated effective masses to other COFs is very useful. If possible, pls include more details of the computational methods used in each instance, as these can affect the m^* values.
- 3) The reported calculations of the effective mass/mobility may require more analysis/discussion. These COFs have two dissimilar in-plane conjugation directions and the calculated values for both are reported in the SI Table 2. The observed trends should be discussed. For example, why does the ratio of m^* along a/b directions changes from 1.8 for COF-0 to 1.1 for COF-1, and then to 4.1 for COF-2? The latter, presumably, relates to the change of the dihedral angle with the Th pendant, but it's harder to see the reason for the higher calculated charge mobility for the hexyl/methyl substituted COF-1 compared to dimethyl-substituted COF-0. I think a deeper analysis of the torsion angles in all calculated structures is needed.
- 4) On a related point, the effective mass/mobility in the out-of-plane direction must be extremely sensitive to the exact stacking mode. What exact stacking mode (offset) was used in calculations? Was it the same for all three COFs?
- 4) The paper explains a smaller (100) diffraction angle in COF-2 vs COF-1 by the lattice expansion. However, the implied difference on the unit cell is $>3\text{\AA}$, which seems large for such structure. On the other hand, the (010) peaks in the reported lattice [i.e., along b direction] should appear at a lower angle but these are not mentioned in either of the COF. The authors should support their assignment by comparing the experimental and simulated PXRDs for both structures. I don't expect a perfect match given the possible dihedral angle disorder along (b) direction, but it would still be useful.
- 5) It would be helpful to specify that the HETCOR correlation of the vinylene hydrogen refers to the R1 chain (not just generic "alkyl chain")
- 6) While the reported increase of conductivity upon n-doping does support the notion of electron transport in these COFs (which is not obvious for such structures), the actual conductivity seems very low. Could the authors comment what limits the conductivity of the doped COFs? Could the radical-anions be chemically unstable? What are the precedents for n-doping of DPP-based linear conjugated polymers? Also, have the authors checked p-doping?
- 7) On a minor point, the synthesis of the model compound 1 is said to be performed in a flask, in MeOH/toluene mixture at 120°C. Was the flask pressurized to achieve such temperature? (if so, a safety note is due).

Reviewer #2

(Remarks to the Author)

Reviewer #3

(Remarks to the Author)

Feng et al. report a comprehensive study about the design and synthesis of two novel electron-deficient 2D covalent organic frameworks (COFs), the so-called 2D poly(arylene vinylene) COFs, and their host-guest interaction with sulfur and sulfides. The developed bipyrazine-based electron-deficient COFs are crystalline and chemically robust. Combining different spectroscopy methods and theoretical calculations, the authors demonstrate that the abundant N sites in bipyrazine boost the electron affinity of COFs, which helps to stabilize the guest sulfur/polysulfide (Li₂S₆) molecules with facilitated intermolecular charge transfer in the porous channels of COFs. They also study the potential of sulfur encapsulated by COFs as electrode materials in Li-sulfur batteries and achieve a high performance. Overall, the result is interesting. I recommend its publication after addressing the following issues:

1. It's important that the authors synthesized model compound to explore the condensation reaction efficiency. A related reaction yield needs to be provided in Scheme S1.
2. The shown result in Figure S1 about tuning the electron deficiency by varying N density is interesting. Can the authors provide a more detailed discussion?
3. Measurement method for Figure 4e has to be provided with more details, for example, the treatment temperature and time of 2D PAVs by 0.1 M Li₂S₆ solution, measurement in glovebox or ambient condition, and so on.
4. What are the specific advantages of using 2D PAVs as hosts compared to other porous materials?
5. It would be beneficial to see what guideline can the authors provide to the future development of electron-deficient 2D PAVs.

Version 2:

Reviewer comments:

Reviewer #1

(Remarks to the Author)

The authors undertook a significant effort to revise the manuscript and carefully addressed all comments from both reviewers. The paper expands the current state-of-art in the design of 2D conjugated COF, and I am happy to recommend accepting this manuscript.

My only (minor) final remark concerns the new ¹H NMR integration data, which show 1.57% CHO in DPP-1 and 1.01% in DPP-2 polymers. If my calculation is correct, the repeat units (BDT + 2DPP) contain 46 (for DPP-1) and 86 (for DPP-2) protons, which implies that there are 2 unreacted aldehyde groups remaining per ~ 3 BDT nodes in the lattice. A similar high level of defects is observed for many reported C=C linked COF (even though most authors don't bother quantifying it), and it seems very plausible that such structural imperfection is the main reason for the inferior long-range transport in COFs compared to structurally similar linear conjugated polymer (where the defect density is 1-2 orders of magnitude lower). I suggest that the author acknowledge this point in the revision. No further review is required.

Reviewer #2

(Remarks to the Author)

The revised manuscript contributed by Zhao and coworkers appears carefully updated, with additional data and explanations that clearly address the concerns and comments of both reviewers. I believe that, with the revisions, the value of the work reporting the record-high mobility of 2D-COF is enhanced. There must be no additional description/data needed in the main text. Personally, it is interesting to note that, based on the integral ratio of aldehyde proton signals reported in Supplementary Figures 21 and 22, the ratio of the unreacted edges can be estimated, which will be a change to estimate the polymerization ratio and then the molecular weight of the 2D-COF. In general, the evaluation of the molecular weight of COF can not be done, and thus, if the molecular weight is estimated in this way, it adds an additional value to the present work from the viewpoint of synthetic chemistry of COF.

Detailed Responses to the Comments of the Reviewers

Reviewer 1:

General comments: X. Feng and co-workers report two new highly conjugated 2D COFs demonstrating an exceptionally small band-gap (1.0 eV) and high intrinsic electron mobility (as shown by DFT and THz measurements). The molecular design takes advantage of the 2D-conjugated building block TBDT-1 explored by the authors in their previous work (refs. 18, 32) and the recently reported Aldol polymerization with dimethyl-DPP (Z. Zhang et al., ACIE 2025, e202417805; this paper should be mentioned in the Introduction). While the low band-gap of DPP-linked conjugated polymers is not itself surprising based on the 1D conjugated polymers literature, the very high charge mobility (especially for electrons) is nonetheless remarkable. I suspect that the high polymerization temperature (200 °C) might limit applications of these COF (due to expected high defect density and difficulty of growing films), but the reported materials still stand out as some of the most promising non-fused semiconducting 2D polymers reported to date. I recommend accepting the paper after a revision of the following points.

Response: We thank reviewer 1 for the positive and insightful comments. In light of your valuable suggestions, we have carefully revised our manuscript.

Following your suggestion, we have revised the introduction with suggested reference on the top of Page 4: *“Very recently, the incorporation of purely planar diketopyrrolopyrrole (DPP) unit, serving as a strongly electron-withdrawing building block, into honeycomb 2D PAVs has markedly enhanced π -conjugation, resulting in narrow optical band gaps.^{24, 25} We envision that extending this design strategy to DPP-based donor-acceptor 2D PAVs in a tetragonal lattice could overcome the cross-conjugation limitations of honeycomb architectures, leading to exceptional charge carrier mobility—an area that remains largely unexplored”*.

Ref. 24: Z. Zhang et al. Diketopyrrolopyrrole-Activated Dynamic Condensation Approach to Narrow-Band Gap Vinylene-Linked Covalent Organic Frameworks. *Angew. Chem. Int. Ed.* 2025, 64, e202417805.

Comment 1: The paper is a bit difficult to follow because the structures of the reported COFs as well as of the models M1-M7 are not shown in the manuscript and have to be guessed from

the provided chemical names (or be dug from the SI). Overall, I feel that too much of essential information is delegated to the SI. On a minor point, I encourage the authors to move the figures from the end of the paper to within the text, to facilitate the reviewer's job.

Response: Thank you for your valuable comment. Following your suggestion, we have moved the chemical structures of M1-M7 from SI to the main text, as shown in Fig. 1b. Meanwhile, the figures are now moved from the end of the paper to within the text to facilitate the review.

Fig. 1 | Molecular design of DPP-based donor-acceptor 2D CPs. (a) Representative structural models of TBDT/DPP-based 2D CPs and model compounds M1–M7. The two strands form along the benzodithiophene (yellow balls) and thienyl (green balls) sites, respectively. (b) Chemical structures of the model compounds M1–M7. (c) HOMO-LUMO

energy levels and gaps of **M1–M7**. **(d)** Molecular frontier orbitals and side-view geometry of **M7**.

Comment 2: The claim of the lowest E_g (among 2D-CPs) should be supported by SI tables summarizing the current state-of-art on that matter.

Response: Thank you for your valuable suggestions. In response, we have summarized the optical absorption maxima of representative 2D CPs in **Supplementary Table 7**. Among them, **2DPAV-TBDT-DPP-1** and **2DPAV-TBDT-DPP-2** display the longest absorption maxima (910 nm and 837 nm, respectively) and the smallest optical band gaps (1.0 eV and 1.2 eV).

Supplementary Table 7. Comparison of band gaps or absorption maxima of the reported 2D CPs.

Sample	Absorption Maxima (nm) ^a	Optical Band Gap (eV) ^b	Reference
2D CPs without DPP units			
TPB-TFB COF	~410	2.6	J. Am. Chem. Soc. 2022, 144, 7489–7496
c-HBC-COF	~500	2.18	J. Am. Chem. Soc. 2022, 144, 5042–5050
COF-Nap	~450	2.28	J. Am. Chem. Soc. 2023, 145, 26871–26882
sp ² c-COF	~500	2.05	Nat. Commun. 2018, 9, 4143
V-2D-COF-W1	~450	2.18	Angew. Chem. Int. Ed. 2022, 61, e202209762
V-2D-COF-W3	~450	2.26	Angew. Chem. Int. Ed. 2022, 61, e202209762
V-2D-COF-W4	~450	2.21	Angew. Chem. Int. Ed. 2022, 61, e202209762
2DPAV-BDT-BP	~500	1.9	Angew. Chem. Int. Ed. 2023, 62, e202305978
2DPAV-BDT-BT	~550	1.62	Angew. Chem. Int. Ed. 2023, 62, e202305978

2DPAV-TBDT-IT	~890	1.15	Angew. Chem. Int. Ed. 2025, e202504302
2DCP-CuPc	~700	1.28	Nat. Mater. 2023, 22, 880–887
2DCP-NiPc	~700	1.33	Nat. Mater. 2023, 22, 880–887
COF-Az	~800	1.37	J. Am. Chem. Soc. 2023, 145, 26871–26882
2D CPs containing DPP units			
g-COF-DPP-1	~710	1.08	Angew. Chem. Int. Ed. 2025, 64, e202417805
g-COF-DPP-2	~690	1.02	Angew. Chem. Int. Ed. 2025, 64, e202417805
DPP-TAPP-COF	~670	-	Angew. Chem. Int. Ed. 2018, 57, 846–850
TpDPP-Py COFs	~750	1.38	Nat. Commun. 2024, 15, 4856
DPP-Py COFs	~600	-	Nat. Commun. 2024, 15, 4856
DPP-TPP-COF	~673	1.63	Small 2024, 2402993
DPP-TBB-COF	~666	1.64	Small 2024, 2402993
2DPAV-TBDT-DPP-1	~910	1.0	This work
2DPAV-TBDT-DPP-2	~837	1.2	This work

^aEstimated from literature. ^bExtracted from the literature.

Comment 3: The SI table 4 comparing the calculated effective masses to other COFs is very useful. If possible, pls include more details of the computational methods used in each instance, as these can affect the m^* values.

Response: Thank you for your suggestion. We have provided detailed information on the computational methods used to determine the effective mass values for each case presented in **Supplementary Table 4**. These details, extracted from the original reports, are summarized and given in **Supplementary Table 5**.

In general, the effective mass tensor of holes (m_h^*) and electrons (m_e^*) are obtained by the following relation:

$$\frac{1}{m^*} = \frac{1}{\hbar^2} \left(\frac{\partial^2 E_n(\vec{k})}{\partial k_i \partial k_j} \right) \quad (i, j = x, y, z)$$

Where \hbar is the reduced Planck constant; x , y , and z denote the direction in k -space and $E_n(\vec{k})$ is the dispersion relation of n -th electronic band; $\frac{\partial^2 E_n(\vec{k})}{\partial k_i \partial k_j}$ is the second derivative of the CBMs or VBMs with respect to wave vector k (can be understood as the curvature by fitting to their DFT-calculated conduction band minimums (CBMs) or valence band maximums (VBMs)).

Since \hbar is a constant, the precision of the effective mass calculation primarily depends on the accuracy of the band structure and the fitting approach used to determine the curvature. When available, the specific band-structure calculation methods and fitting procedures employed in each case are listed in **Supplementary Table 5**. Correspondingly, the effective mass values used for carrier mobility calculations are also summarized therein.

The band structure calculation methods vary among references — for instance, PBE functional (Refs. 10, 11, 12, 13, 16, 17, 18, and 27), third-order DFTB with the 3ob-3-1 parameter set (Ref. 14), B3LYP/6-311G(d,p) level (Ref. 19), and HSE06 (Refs. 15, 27, 20, 21, 22, 23, and 24). In this study, we employed the PBE functional, which is widely adopted for similar systems and provides a good balance between computational efficiency and accuracy in describing electronic structures.

Likewise, different fitting methods were applied, such as parabolic fitting and the finite difference method using a five-point stencil. In our calculations, we applied the parabolic fitting process, which is the most commonly used approach for extracting effective masses near the band extrema and ensures reliable comparison with literature data.

In most cases, the reduced effective mass is further used to estimate carrier mobility by considering the contributions from both electrons and holes. Consistently, we also adopted the reduced effective mass in our mobility calculations to enable direct and meaningful comparison with previously reported results.

Supplementary Table 5. Details of the effective mass calculations for the examples summarized above.

Ref.	Calculation method
General	The effective mass tensor of holes (m_h^*) and electrons (m_e^*) are obtained by the following relation: $\frac{1}{m^*} = \frac{1}{\hbar^2} \left(\frac{\partial^2 E_n(\vec{k})}{\partial k_i \partial k_j} \right) \quad (i, j = x, y, z)$ Where \hbar is the reduced Planck constant; x, y, and z denote the direction in k-space and $E_n(\vec{k})$ is the dispersion relation of n-th electronic band; $\frac{\partial^2 E_n(\vec{k})}{\partial k_i \partial k_j}$ is the second derivative of the CBMs or VBMs with respect to wave vector k (can be simply understood as the curvature in the band dispersion)
Ref. 10	Band structure calculation: Generalized gradient approximation (GGA) of the exchange-correlation energy in the form of Perdew-Burke-Ernzerhof (PBE) was applied Fitting method: Parabolic fitting of the VBM and CBM using SUMO Python toolkit or manually in the case of nearly degenerated electronic states Effective mass used for mobility calculation: electron-hole reduced effective mass (m^*) considering averaged effective masses for electrons ($m_{e(avg)}^*$) and holes ($m_{h(avg)}^*$) $\frac{1}{m^*} = \frac{1}{m_{e(avg)}^*} + \frac{1}{m_{h(avg)}^*}$
Ref. 11	Band structure calculation: Exchange-correlation functional was treated at GGA level in the form of PBE Fitting method: the finite difference method on a five-point stencil Effective mass used for mobility calculation: only considered the component m_{zz}^* mobility
Ref. 12	Band structure calculation: PBE-D3/DZ Fitting method: the finite difference method on a five-point stencil Effective mass used for mobility calculation: the reduced electron-hole effective mass (m^*) $\frac{1}{m^*} = \frac{1}{m_e} + \frac{1}{m_h}$
Ref. 13	Band structure calculation: DFT calculations were performed using the Vienna ab-initio simulation package (VASP). A GGA in the form of PBE is used for the exchange and correlation functional Fitting method: Parabolic fitting

	Effective mass used for mobility calculation: To take into account the effective mass contribution from electrons and holes for mobility, they further use the reduced effective mass of charge carriers m^* following: $\frac{1}{m^*} = \frac{1}{m_e^*} + \frac{1}{m_h^*}$
Ref. 14	Band structure calculation: Geometries were calculated using Density Functional Tight Binding (DFTB) as implemented in DFTB+ version 20.1. Following geometry optimization, the band structure, Density of States (DOS) and effective mass were calculated using 3rd order DFTB and the 3ob-3-1 parameter set Fitting method: Not mentioned Effective mass used for mobility calculation: 3rd order DFTB and the 3ob-3-1 parameter set
Ref. 15	Band structure calculation: Calculation was carried out under k-point grid of $3 \times 3 \times 6$, in implement with the TIER1 basis set and tight integration grid. Herd–Scuseria–Ernzerhof hybrid functional (HSE06) was used with 25% Hartree–Fock exchange and a screening parameter of 0.11 bohr^{-1}. Many-body method was adopted to correct the long-range van der Waals dispersion Fitting method: Not mentioned Effective mass used for mobility calculation: the reduced electron-hole effective mass calculated by: $\frac{1}{m^*} = \frac{1}{m_e} + \frac{1}{m_h}$
Ref. 16	The same as Ref. 10
Ref. 17	The same as Ref. 11
Ref. 18	Band structure calculation: GGA of the exchange-correlation energy in the form of PBE was applied Fitting method: Not mentioned Effective mass used for mobility calculation: An electron-hole averaged effective mass
Ref. 19	Band structure calculation: geometries of monomer were optimized at the B3LYP/6-311G(d,p) level Fitting method: Not mentioned Effective mass used for mobility calculation: not applied
Ref. 20	Band structure calculation: DFT level of theory with the HSE06 functional and 6-31G(d) basis set Fitting method: Not mentioned

	Effective mass used for mobility calculation: From the band dispersion, they computed the reduced mass around the VBM and CBM, as well as the VBM and CBM orbitals. Furthermore, they estimated the effective reduced mass of charge carriers m^*, by considering the averaged values for both charges, with the equation: $\frac{1}{m^*} = \frac{1}{m_e^*} + \frac{1}{m_h^*}$
Ref. 21	Band structure calculation: All computations have been performed with the Gaussian16 software, considering the DFT level of theory and the range-separated screened HSE hybrid functional with the Pople's 631G(d) basis set Fitting method: Not mentioned Effective mass used for mobility calculation: As both the electron and hole contribute to the photoconductivity, they approximate the mass by the averaged value of both charges and use the reduced mass of charge carriers calculated by: $\frac{1}{m^*} = \frac{1}{m_e^*} + \frac{1}{m_h^*}$ The effective masses of the electron and the hole have been calculated by DFT calculation performed on the optimized cGNR structure
Ref. 22	The same as Ref. 21
Ref.23	Band structure calculation: DFT level of theory with the HSE functional and 6-31G(d) basis set, using the Gaussian16 suite of programs. For the infinite polymer, periodic boundary conditions (PBC) were applied Fitting method: Not mentioned Effective mass used for mobility calculation: Not mentioned
Ref.24	Band structure calculation: The electronic structure was calculated at the DFT level using the screened exchange hybrid exchange–correlation functional HSE06 and the standard 6-31G* basis set Fitting method: Not mentioned Effective mass used for mobility calculation: Not mentioned
Ref.25	Band structure calculation: Not mentioned Fitting method: Not mentioned Effective mass used for mobility calculation: Use the reported effective mass
Ref.26	Not mentioned
Ref.27	Band structure calculation: The PBE exchange-correlation functional and the projector-augmented wave (PAW) method were employed for structural optimization Fitting method: Not mentioned Effective mass used for mobility calculation: use effective mass of charge carriers in the 5-layer BP obtained from the literature

Ref. 28–35 TRMC, rather than THz technique, doesn't need effective mass values

This work **Band structure calculation:** The electronic properties of the 2D PAVs were calculated using density functional theory (DFT) with the Vienna Ab Initio Simulation Package (VASP 5.4.4), implementing the projector-augmented wave (PAW) method and the PBE functional

Fitting method: Parabolic fitting

Effective mass used for mobility calculation: In our studies, we do not distinguish the contribution of electrons and holes. The inferred scattering time is the averaged value with the contribution of both types of charge carriers. Therefore, we take the reduced electron-hole mass by taking the contribution of both electrons and holes into account following:

$$\frac{1}{m_{e-h}^*} = \frac{1}{m_e^*} + \frac{1}{m_h^*}$$

to infer the final mobility value. In which, the m_e^* and m_h^* represent electron and hole effective masses, respectively

Comment 4: The reported calculations of the effective mass/mobility may require more analysis/discussion. These COFs have two dissimilar in-plane conjugation directions and the calculated values for both are reported in the SI Table 3. The observed trends should be discussed. For example, why does the ratio of m^* along a/b directions changes from 1.8 for COF-0 to 1.1 for COF-1, and then to 4.1 for COF-2? The latter, presumably, relates to the change of the dihedral angle with the Th pendant, but it's harder to see the reason for the higher calculated charge mobility for the hexyl/methyl substituted COF-1 compared to dimethyl-substituted COF-0. I think a deeper analysis of the torsion angles in all calculated structures is needed.

Response: Thanks for your insightful suggestions. We have now analyzed the torsion angles in all calculated structures (2DPAV-TBDT-DPP-0, 2DPAV-TBDT-DPP-1 and 2DPAV-TBDT-DPP-2). In each structure, the torsion angle under analysis is highlighted with green spheres, as shown in **Supplementary Figure 16** and summarized in **Table R1**.

Supplementary Figure 16. Analysis of the torsion angles of optimized models of **2DPAV-TBDT-DPP-0** (top), **2DPAV-TBDT-DPP-1** (middle), and **2DPAV-TBDT-DPP-2** (bottom).

Table R1. Torsion angles of layer-stacked **2DPAV-TBDT-DPP-0/1/2**.

	Torsion angles along b			Torsion angles along a		
	Angle 1	Angle 2	Average	Angle 3	Angle 4	Average
2DPAV-TBDT-DPP-0	152.8°	164.6°	21.3°	4.40°	4.56°	4.48°
2DPAV-TBDT-DPP-1	151.8°	164.1°	22°	5.23°	5.22°	5.225°
2DPAV-TBDT-DPP-2	132.4°	134.9°	46.3°	1.83°	3.48°	2.655°

Distinct anisotropy is observed in all three structures. Along the *a* direction, the torsion angles fall within the range of 1.8–5.3°, which are much smaller than those along the *b* direction (15–47°) (see above in **Supplementary Fig. 16** and **Table R1**). The good planarity along *a* direction

is associated with enhanced π -cloud delocalization, leading to a smaller electron effective mass compared with the b direction.

We further examined the relationship between the ratio of torsion angles (b/a , averaged values) and the ratio of effective masses (b/a). As shown in **Supplementary Figure 17**, the trend in effective mass ratios follows that of the torsion angle ratios. This implies a molecular design principle: high planarity promotes low effective mass, while deviations from planarity induce transport anisotropy. Related discussions are provided in Revised Manuscript on Page 10: “*The effective masses of electrons and holes exhibit a strong correlation with the average torsion angles of the polymer backbone along the a and b directions. The variation in their mass ratios closely follows the trend of the corresponding torsion angle ratios (Supplementary Figs. 16 and 17), suggesting that higher backbone planarity leads to lower effective mass, whereas deviations from planarity induce pronounced charge-transport anisotropy.*”

Supplementary Figure 17. Analysis of the ratio of effective masses and the ratio of averaged torsion angles along a and b directions.

We are sorry for the confusion that “*The latter, presumably, relates to the change of the dihedral angle with the Th pendant, but it’s harder to see the reason for the higher calculated charge mobility for the hexyl/methyl substituted COF-1 compared to dimethyl-substituted COF-0*”. In fact, we have only predicted the charge carrier mobility for “COF-0” due to limited

computational resources. However, from **Supplementary Table 3**, we indeed show that “COF-1” has a smaller electron effective mass than “COF-0” (hole effective mass remains similar). The smaller electron effective mass of “COF-1” is due to the stronger van der Waals interactions and more favorable inclined stacking induced by the longer alkyl side chain in COF-1, which enhances interlayer coupling and π - π overlap, thereby reducing the electron effective mass compared with that of COF-0. This observation aligns well with previous studies demonstrating that alkyl side chain modification can substantially vary interlayer van der Waals interactions and stacking geometry of framework materials, thereby tuning π - π overlap, electronic coupling, and charge transport. In particular, medium-length alkyl chains have been shown to reinforce interlayer interactions and improve charge mobility (*J. Am. Chem. Soc.* 2023, 145, 21798; *Chem. Mater.* 2014, 26, 594.).

Comment 5: On a related point, the effective mass/mobility in the out-of-plane direction must be extremely sensitive to the exact stacking mode. What exact stacking mode (offset) was used in calculations? Was it the same for all three COFs?

Response: We agree with the reviewer that the out-of-plane effective mass and mobility are highly sensitive to the stacking mode. The stacking modes are identical for all three COFs in this work, specifically is inclined AA (AA') stacking, as shown in **Supplementary Figure 12 (2DPAV-TBDT-DPP-0)**, **13 (2DPAV-TBDT-DPP-1)**, and **14 (2DPAV-TBDT-DPP-2)**. The interlayer offsets for these COFs are 0.8, 0.7, and 1.2 Å for **2DPAV-TBDT-DPP-0**, **2DPAV-TBDT-DPP-1**, and **2DPAV-TBDT-DPP-2**, respectively. Such AA' stacking configuration is consistent with those reported in previous COF studies (*Chem. Sci.* 2020, 11, 12647). The weak van der Waals interactions governing interlayer stacking may lead to increased disorder at room temperature, potentially affecting interlayer charge transport (*Chem. Mater.* 2022, 34, 2376).

Supplementary Figure 12. Models of 2DPAV-TBDT-DPP-0.

Supplementary Figure 13. Models of 2DPAV-TBDT-DPP-1.

Supplementary Figure 14. Models of 2DPAV-TBDT-DPP-2.

It is noted that the intralayer effective mass is several orders of magnitude smaller than that in the out-of-plane direction due to strong in-plane π -conjugation. Consequently, charge carrier transport along the in-plane direction is significantly enhanced in our current case, dominating the overall transport behavior (**Supplementary Table 8**). Given this, the in-plane transport properties are the primary focus of this work.

Comment 6: The paper explains a smaller (100) diffraction angle in COF-2 vs COF-1 by the lattice expansion. However, the implied difference on the unit cell is $> 3\text{\AA}$, which seems large for such structure. On the other hand, the (010) peaks in the reported lattice [i.e., along b direction] should appear at a lower angle but these are not mentioned in either of the COF. The authors should support their assignment by comparing the experimental and simulated pXRDs for both structures. I don't expect a perfect match given the possible dihedral angle disorder along (b) direction, but it would still be useful.

Response: We appreciate the reviewer's insightful comments regarding the lattice differences between **2DPAV-TBDT-DPP-1** and **2DPAV-TBDT-DPP-2** and the assignment of diffraction peaks. In response, we clarify the observed lattice expansion and provide additional details to support our peak assignments on Page 11, as shown in **Fig. 3b**. *“It displays a similar pXRD pattern to 2DPAV-TBDT-DPP-1, but with a shift of the predominant diffraction from the (100) to the (010) plane at a lower angle of 4.31° , corresponding to a larger b-lattice parameter than a. 2DPAV-TBDT-DPP-2, with its longer hexyl (C6) chains compared to the methyl (C1)*

chains in 2DPAV-TBDT-DPP-1, exhibits a slight lattice expansion, particularly along the c-direction (+0.3 Å), with smaller changes along the a-direction (+0.1 Å) and b-direction (-0.3 Å) (see details in Supplementary Fig. 13 and 14). These changes result in little shifts in diffraction angles for the (100) and (010) peaks. We note that interlayer sliding in these COFs can further influence the positions of these peaks, contributing to the observed diffraction patterns”.

We fully agree with the reviewer that the unit cell differences along the *a*- and *b*-directions are substantial, measuring 4.4 Å and 3.9 Å for **2DPAV-TBDT-DPP-1** and **2DPAV-TBDT-DPP-2**, respectively. We attribute these differences to the asymmetric geometry of the TBDT node, which introduces anisotropy in the lattice. Similar behavior is observed in other anisotropic systems, such as multi-component COFs (*Nat. Commun.* 2016, 7, 12325; *J. Am. Chem. Soc.* 2019, 141, 18004).

Following the reviewer’s suggestion, we have included a detailed comparison of experimental and simulated pXRD patterns for both **2DPAV-TBDT-DPP-1** and **2DPAV-TBDT-DPP-2** to support our peak assignments. We acknowledge that a perfect match between experimental and simulated pXRD patterns is challenging due to the inherent flexibility of the COF frameworks and alkyl chains, as well as potential dihedral angle disorder along the *b*-direction, as noted by the reviewer. Additionally, the limited resolution of the current pXRD data constrains precise peak assignments. Nevertheless, the provided comparisons strengthen the reliability of our structural analysis.

Fig. 3 | Synthesis and characterization of TBDT/DPP-based crystalline 2D PAVs. (a) Schematic synthesis. **(b)** Experimental, Pawley-refined and simulated pXRD patterns as well as the difference plots of **2DPAV-TBDT-DPP-1** (top) and **2DPAV-TBDT-DPP-2** (bottom), respectively. **(c)** Solid-state ¹³C CP MAS NMR spectra of **2DPAV-TBDT-DPP-1**. The liquid-state NMR spectra of the monomers are shown for comparison. **(d)** ¹H-¹³C HETCOR spectra of **2DPAV-TBDT-DPP-1**. **(e,f)** Top and side views of the structure model. **(g)** HR-TEM image and the inset for the SAED pattern. **(h, i)** SEM images of **2DPAV-TBDT-DPP-1** (**h**) and **2DPAV-TBDT-DPP-2** (**i**).

Comment 7: It would be helpful to specify that the HETCOR correlation of the vinylene hydrogen refers to the R₁ chain (not just generic “alkyl chain”).

Response: Thanks for your suggestion. We now specify the potential correlated carbons with the hydrogen bonding formed by the vinylene hydrogen. We have added additional discussion to the revised manuscript on Pages 13 and 14: “The proton signal corresponding to the vinylene linkage appears at 9 ppm due to hydrogen bonding and shows correlation with the spatially closest carbon (marked by a green circle) of the flexible aliphatic chain (blue dashed line circled area in Fig. 3d)”.

Comment 8: While the reported increase of conductivity upon n-doping does support the notion of electron transport in these COFs (which is not obvious for such structures), the actual conductivity seems very low. Could the authors comment what limits the conductivity of the doped COFs? Could the radical-anions be chemically unstable? What are the precedents for n-doping of DPP-based linear conjugated polymers? Also, have the authors checked p-doping?

Response: Thank you for raising this point. The relatively low conductivity after *n*-doping indeed touches upon a central challenge in this field. The factors limiting the conductivity in doped COF materials are multifaceted, and we will explain them by combining precedents from the literature with the specific characteristics of our system.

In contrast to the more established *n*-doping of DPP-based linear conjugated polymers, the performance of *n*-doped 2D COFs remains underdeveloped, as summarized in **Supplementary Table 9**. In general, high electrical conductivity of linear conjugated polymers is achieved through LUMO energy level modulation (*J. Am. Chem. Soc.* 2019, 141, 20215; *Angew. Chem. Int. Ed.* 2023, 62, e202216049; *J. Mater. Chem. C* 2022, 10, 2718), miscibility tuning (*Acc. Chem. Res.* 2025, 58, 9, 1496; *Adv. Mater.* 2018, 30, 1704630), polymer geometry optimization (*Nat. Commun.* 2021, 12, 5723), and dopants design and screening (*Nature* 2025, 642, 599; *Nat. Commun.* 2020, 11, 3292).

Supplementary Table 9. Summary of the *n*-doping behavior of DPP-based linear conjugated polymers and 2D COFs.

Linear conjugated polymers			
Sample	n -dopants	Conductivity (S/cm)	Reference
P(TDPP-CT2)	N -DMBI	0.39	J. Am. Chem. Soc. 2019, 141, 20215–20221

P(PzDPP-CT2)	N -DMBI	8.4	J. Am. Chem. Soc. 2019, 141, 20215–20221
P(PzDPP-2FT)	CoCp ₂	129	Nat. Commun. 2021, 12, 5723
P(PzDPP-2FT)	N -DMBI	43.3	Nat. Commun. 2021, 12, 5723
P(BTP-DPP)	(RuCp*mes) ₂	0.45	Chem. Mater. 2017, 29, 22, 9742–9750
pDFSe	N -DMBI	62.6	Angew. Chem. Int. Ed. 2024, 63, e202409018
ThDPP-CNBTz	N -DMBI	50.6	Angew. Chem. Int. Ed. 2024, 63, e202402642
PTz-5-DPP	N -DMBI	8	Angew. Chem. Int. Ed. 2023, 62, e202219262
2D COFs			
DPP-TPP-COF	TDAE	5.1×10^{-5}	Small 2024, 2402993
DPP-TBB-COF	TDAE	1.2×10^{-5}	Small 2024, 2402993

In contrast, limited efforts are put into 2D COFs design. More investigation and understanding are still required. Considering big challenges in achieving efficient *n*-doping, although primary, our work provides valuable experience in designing 2D COFs that are suitable for *n*-doping.

The low conductivity of **2DPAV-TBDT-DPP-1** and **2DPAV-TBDT-DPP-2** can be attributed to several factors. First, the chemical instability of radical anions is critical. Our design employs strong electron-deficient units, maintaining doped-state stability demands inert conditions, complicating conductivity measurements. However, the introduced radical anions were completely annihilated after exposure to air for 5 minutes (**Figure R1**). Second, doping efficiency is limited. Our doping and measurement in its powder form is not as efficient as in films for conventional linear conjugated polymers. In addition, the relatively small pore size restricts dopant accessibility under the current condition, thereby diminishing their doping efficiency. In this work, we conducted preliminary experiments to demonstrate that the two 2D PAVs can be *n*-doped. Moving forward, achieving efficient *n*-doping and high electrical conductivity will require our more systematic investigation and a deeper understanding in the future.

Figure R1. Stability of n-doping of 2DPAV-TBDT-DPP-1.

Following your suggestion, we also checked *p*-doping with our 2D PAVs and found that the developed 2D PAVs can be doped by I₂ vapor. Upon doping with I₂ vapor for 5 min, we observed a current increase from 10⁻⁵ A (pristine) to 10⁻⁴ A, suggesting an enhanced conductivity (**Figure R2**). However, the *p*-doped materials did not show good ohmic contact.

Figure R2. In-situ I₂ doping (left) of 2DPAV-TBDT-DPP-1 (middle) and 2DPAV-TBDT-DPP-2 (right).

Comment 9: On a minor point, the synthesis of the model compound 1 is said to be performed in a flask, in MeOH/toluene mixture at 120 °C. Was the flask pressurized to achieve such temperature? (if so, a safety note is due).

Response: Thank you for your suggestions. Indeed, the container is pressurized to achieve high-pressure environment required for the synthesis. We have revised the synthesis of model compound 1 as “**DPP-2** (20 mg, 0.06 mmol), thiophene-2-carbaldehyde (*Th-CHO*) (14.84 mg,

0.132 mmol), and sodium benzoate (1 mg) were added into a Synthware thick wall pressure bottle fitted with toluene/methanol (5:1) under argon protection” in Supplementary Information.

Reviewer 2:

The manuscript reported by Zhao and coworkers describes the synthesis and characterization of DPP-incorporated 2D COF for use in electronic materials. One of the key features of the work is the record high electron mobility of the materials developed, i.e., 2DPAV-TBDT-DPP-1 and 2DPAV-TBDTDPP-2. The mobility values are the highest not only for similar 2D-COFs in the field, but also for related 1D conjugated polymers consisting of the DPP units... With revisions and additions concerning the above points, the manuscript could be re-reviewed.

Response: We thank reviewer 2 for the positive and insightful comments. In response to your valuable suggestions, we have carefully revised the manuscript to include additional characterizations and analyses.

Comment 1: Thus, provided that the performances are real, the present results are quite appealing and noteworthy. In this regard, evaluating mobility is key, as discussed below. The mobility was estimated from the equation mentioned in the main text, line 221, where the key parameters are τ_s , c , and m^* . As far as I understand, the former two are extracted from OPTP measurements, and m^* appears to be obtained from theoretical calculations for the band structures. My concern is that the values of m^* obtained from calculations based on the ideal periodic boundary model can easily overestimate m^* , which can be extracted from the curvature in the band dispersion. This may not be realistic for the actual material in the present case. This is due to the low structural perfection of 2DPAV-TBDT-DPP-1 and 2DPAV-TBDT-DPP-2, as indicated by the ^1H and ^{13}C NMR spectra reported in Supplementary Figures 19 and 20. In Supplementary Figure 19, a broad band is observed at around 10 ppm, which can be assigned to the aldehyde moiety of TBDT-1 (and related intermediates). The same is true for the ^{13}C NMR, in which peaks around 180 ppm can also be assigned to the aldehyde moiety. These strongly imply that the structural perfection of 2DPAV-TBDTDPP-1 and 2DPAV-TBDT-DPP-2 is not that high, and then the m^* s, and accordingly the mobility values, should be carefully reconsidered.

Response 1: Thank you for your constructive comments. Indeed, before achieving perfect single crystals, band structure simulation based on the obtained polycrystalline materials is the typical method to derive effective mass and further to obtain the charge carrier mobility. Regarding this, we have provided detailed information on the computational methods used to

determine the effective mass values in the literature. These details, extracted from the original reports, are summarized and given in **Supplementary Table 5**. The specific calculation procedures employed in this study are also described in the last column of **Supplementary Table 5** and further detailed in **Supplementary Information** (see Pages S5 and S38).

To address your concern about structural perfection, we further evaluate the chemical and crystalline structures of the as-synthesized **2DPAV-TBDT-DPP-1** and **2DPAV-TBDT-DPP-2**.

As you mentioned, there are still unreacted -CHO groups as illustrated by ss-¹H and ss-¹³C NMR spectra. We quantified the unreacted -CHO groups *via* proton ratio analysis of ¹H NMR spectrum (**Supplementary Figures 21 and 22**), revealing minimal remaining aldehyde protons of 1.57% for **2DPAV-TBDT-DPP-1** and 1.01% for **2DPAV-TBDT-DPP-2**. These residual aldehyde groups likely reside at the edges of the 2D PAV, a common phenomenon observed in COF materials, for example, imine-linked (*Nat. Commun.* 2022, 13, 1370; *Nat. Commun.* 2022, 13, 1411) and imidazole-linked COFs (*Nat. Commun.* 2020, 11, 2021). We have added additional discussion to the Revised Manuscript on Page 12, “A small signal at 180 ppm can be attributed to the unreacted -CHO groups that may remain at the edge of the **2DPAV-TBDT-DPP-1** and **2DPAV-TBDT-DPP-2** particles (Supplementary Figs. 21 and 22, the ratios of aldehyde protons are determined to be 1.57% and 1.01%, respectively)”.

Supplementary Figure 21. Measured ss-depth ¹H NMR spectrum (left) of **2DPAV-TBDT-DPP-1** and deconvolution⁹ (right) of the spectrum. The ratio of unreacted aldehyde C is determined to be 1.57%.

Supplementary Figure 22. Measured ss-depth ^1H NMR spectrum (left) of **2DPAV-TBDT-DPP-2** and deconvolution⁹ (right) of the spectrum. The ratio of unreacted aldehyde C is determined to be 1.01%.

Regarding the crystalline structure, the average crystal size and long-range ordering of the material can be evaluated from the full width at half-maximum (FWHM) value of the peaks at the lowest angle (*Nat. Commun.* 2021, 12, 1982; *J. Am. Chem. Soc.* 2022, 144, 6594–6603). We summarized and compared the FWHM values with representative and highly crystalline 2D COFs in **Supplementary Table 6**. For 2D COFs bridged by a variety of linkages (imine, vinylene, imide, triazine, quinoline...), the FWHM values of our materials are among the smallest, indicating the high degree of the structural order of the obtained materials.

Supplementary Table 6. Comparison of FWHM_{100} of the representative 2D COFs with different linkages.

Sample	Linkage	FWHM_{100} (degree)	Reference
g-C ₄₀ N ₃ -COF	C=C	0.305	Nat. Commun. 2019, 10, 2467
g-C ₃₁ N ₃ -COF	C=C	0.884	Nat. Commun. 2019, 10, 2467
g-C ₃₇ N ₃ -COF	C=C	0.934	Nat. Commun. 2019, 10, 2467
v-2D-COF-NS1	C=C	0.7	ACS Catal. 2023, 13, 1089–1096
COF-1	Triazine	0.3	Angew. Chem. Int. Ed. 2019, 58, 13753-13757

g-DZPH-COF	C=C	0.19	Angew. Chem. Int. Ed. 2024, 63, e202402446
g-DZTA-COF	C=C	0.26	Angew. Chem. Int. Ed. 2024, 63, e202402446
V-2D-COFs	C=C	0.96-1.03	Angew. Chem. Int. Ed. 2022, 61, e202209762
ivCOF-1-Br	C=C	0.783	Angew. Chem. Int. Ed. 2021, 60, 13614–13620
ivCOF-2-Br	C=C	0.528	Angew. Chem. Int. Ed. 2021, 60, 13614–13620
ivCOF-2-I	C=C	0.635	Angew. Chem. Int. Ed. 2021, 60, 13614–13620
g-COF-DPP-1	C=C	0.428	Angew. Chem. Int. Ed. 2025, 64, e202417805
g-COF-DPP-2	C=C	0.535	Angew. Chem. Int. Ed. 2025, 64, e202417805
NKCOF-28	imide	0.792	Chem. , 2023, 9, 2178–2193
LZU-600	pyrano[4,3- b]pyridine	0.46	J. Am. Chem. Soc. 2022, 144, 6594–6603
LZU-601		0.28	J. Am. Chem. Soc. 2022, 144, 6594–6603
LZU-602		0.50	J. Am. Chem. Soc. 2022, 144, 6594–6603
LZU-603		0.40	J. Am. Chem. Soc. 2022, 144, 6594–6603
TFPPy-PDA-COF	imine	0.38	J. Am. Chem. Soc. 2022, 144, 9624–9633
TPB-DMTP-COF	imine	0.39	Nat. Chem. 2015, 7, 905–912
[S-Py] _{0.17} -TPB-DMTP-COF	imine	0.437	Nat. Chem. 2015, 7, 905–912
sc-COF _{TP-Py}	imine	0.34	Nat. Commun. 2021, 12, 5077
NKCOF-10	C=C	0.81	Nat. Commun. 2021, 12, 1982
QL-COF-2	4-carboxyl-quinoline	1.04	Nat. Commun. 2022,13, 2615
2DPAV-TBDT-DPP-1	C=C	0.39	This work

Meanwhile, we have calculated the mean free path (λ) of charge carriers for the two 2D PAVs under the formula:

$$\lambda = \frac{v_F m^* \mu}{e}$$

in which the mean free path (λ) of charge carriers is a function of the Fermi velocity (v_F), the carrier effective mass (m^*), the mobility (μ), and the elementary charge (e). Using the obtained charge mobility, we estimate the longest migration path for the free charge carriers as 2.36 nm and 8.37 nm for **2DPAV-TBDT-DPP-1** and **2DPAV-TBDT-DPP-2**, respectively.

The average crystallite size of the materials is further estimated using the Scherrer equation:

$$D = \frac{K\lambda}{\beta \cos \theta}$$

where D is the mean size of coherently diffracting domains, K is the shape factor (typically ~ 0.9 , depending on crystallite geometry), λ is the X-ray wavelength (1.5406 Å for Cu K α radiation), β is the FWHM of the diffraction peak (corrected for instrumental broadening and expressed in radians), and θ is the Bragg angle corresponding to the peak position. Using this approach, the average crystal size of **2DPAV-TBDT-DPP-1** was estimated to be ~ 20 nm. It is also worth noting that the Scherrer equation typically underestimates the crystal size of porous framework materials.

Thus, the estimated crystalline domain size of the as-synthesized 2D PAVs is significantly larger than the diffusion length of free charge carriers probed by THz microscopy. Considering that THz spectroscopy primarily probes the intrinsic and short-range charge transport within individual crystalline domains of **2DPAV-TBDT-DPP-1** and **2DPAV-TBDT-DPP-2** (**Figure R3**), the carrier mobilities extracted from THz measurements are therefore considered reasonable. We hope that our results can address your concern.

Figure R3. Demonstration of the free charge carrier transporting scenario in polycrystalline 2DPAV-TBDT-DPP-1.

Comment 2: Furthermore, the enhanced conductivity resulting from doping with N-DMBI is mentioned in the main text, lines 230-233; however, no quantitative evaluation of electrical conductivity is reported in Supplementary Figure 31. More detailed data should be added.

Response 2: We appreciate your valuable suggestions. At this stage, we are unable to provide precise conductivity values, as our measurement method does not allow for precise quantification of the sample layer thickness. The measurements were conducted using a two-probe method with a Keithley 2450 SourceMeter. The suspension was prepared by drop-casting powder samples onto pre-patterned PTFE-supported electrodes. However, this process results in non-continuous and -uniform thickness (in the range of tens to hundreds nm), and despite knowing the channel width, we were unable to determine the exact conductivity (**Figure R4**). Instead, we can estimate the improvement in conductivity before and after doping. Our aim here is merely to demonstrate that the as-synthesized materials can be successfully n-doped. The employed doping and measurement methods in its powder form are not as efficient as in films for conventional linear conjugated polymers.

Figure R4. A representative picture of our setup for I-V curve measurement.

Comment 3: For these reasons, the authors are advised to evaluate the materials more carefully and should provide clear spectrum data, eliminating any doubt about the characterization of the materials. In this sense, in Supplementary Figures 19-22 (21 and 22 are IR spectra), the spectra of “Model compound 1 (mentioned on page S9 in the Supplementary Information)” should be added to compare with the spectra.

Response 3: Following your suggestion, we further compared the FT-IR spectra to investigate the transformation of aldehyde and methyl groups into vinylene moieties. As recommended, we first analyzed the FT-IR spectra of monomers (DPP-2 and Th-CHO) and model compound 1 to gain insight. The corresponding data are provided in **Supplementary Figure 24** (see below). The experimental FT-IR spectra show excellent agreement with the simulated spectra (a correction factor of 0.959 was applied). For DPP-2 monomer (top), the stretching vibrations of the C=O (ketone) and C=C groups appear at 1683 and 1628 cm^{-1} , respectively. In Th-CHO monomer, the C=O (aldehyde) stretching vibration is observed at 1655 cm^{-1} . By contrast, model compound 1 exhibits red-shifted stretching vibrations of the C=O (ketone) and C=C groups at 1663 and 1564 cm^{-1} , respectively, consistent with the effect of extended conjugation. Moreover, the yellow-highlighted bar indicates the C=O stretching region, where the peaks are closely spaced and can only be clearly distinguished in the zoomed-in view. A similar feature is observed in the FT-IR spectra of **2DPAV-TBDT-DPP-1** and **2DPAV-TBDT-DPP-2** (**Supplementary Figures 27 and 28**).

Supplementary Figure 24. Experimental (Exp.) and simulated (Sim.) FT-IR spectra of DPP-2 (top), Th-CHO (middle), and model compound 1 (bottom).

We further analyzed the FT-IR spectra of the model compound 1, **2DPAV-TBDT-DPP-1**, and **2DPAV-TBDT-DPP-2**, which are provided in **Supplementary Figure 25**.

Supplementary Figure 25. FT-IR spectra of model compound 1, **2DPAV-TBDT-DPP-1**, and **2DPAV-TBDT-DPP-2**.

Raman spectra of the above materials are also measured (**Figure R5**). Similar features corresponding to aromatic C=C stretching, CH₃/CH₂ bending, aromatic C-H deformation, and C-S stretching are observed in the model compound 1, **2DPAV-TBDT-DPP-1**, and **2DPAV-TBDT-DPP-2**.

Figure R5. Raman spectra of model compound 1, **2DPAV-TBDT-DPP-1**, and **2DPAV-TBDT-DPP-2**.

Comment 4: Also, the synthetic yield of “Model compound 1” is reported to be 65%, meaning that the efficiency of the basic reaction for constructing the 2D COF is not very high. Considering this and the almost quantitative yields for 2DPAV-TBDT-DPP-1 and 2DPAV-TBDT-DPP-2, the structural characterization of these materials should be conducted with the utmost care.

Response 4: We thank you for this valuable comment. Although the model reaction yield we report here is 65% (performed using sodium benzoate as the catalyst in toluene/methanol (5:1) under argon at 120 °C overnight), the actual 2D polymerization was carried out under energetically more favorable conditions, namely, at an elevated temperature (200 °C for 3 days) and in the solid state. Based on our analysis (see our response to Comment 1 above), these conditions enable efficient 2D polymerization and afford highly crystalline 2D PAVs with minimal structural defects. However, due to the low boiling point of thiophene-2-carbaldehyde, carrying out the model reaction under the same solid-state, high-temperature conditions (200 °C) as those used for the 2D polycondensation is not feasible. Moreover, under other reported conditions, for example, reaction No. 15 (**Supplementary Table 2**) employing L-proline/TA as the catalyst (with yield up to 90%, *Org. Lett.* 2019, 21, 1973–1978), we could not obtain crystalline 2D PAV.

Comment 5: Another comment to the authors is that I was a little bothered by a feeling of strangeness regarding the description of the “Design of DPP-based donor-acceptor polymer backbone (line 83-100)”. They compared M1-M7 with different p-conjugated monomers, including benzene (M1), pyridine (M2), pyradine (M3), furan (M4), thiophene (M5), selenophene (M6), but for the synthesis of 2D polymer with these aromatic units, different synthetic reactions than the present vinyl (M7) unit, like transition metal-catalyzed cross coupling reactions, are necessary. From a synthetic chemistry point of view, the use of the vinyl unit, which is constructed by cross aldol condensation of dimethyl-DPP and aldehyde, followed by dehydration to furnish the vinyl moiety, is only a feasible method to build 2DCOF, as the first aldol condensation is an equilibrium reaction, which allows for to formation of the ideal 2D structure, and then dehydration (non-equilibrium reaction) to fix the 2D-network structure. Thus, the authors are recommended to reconsider this part.

Response 5: Thank you for your insightful comments. We fully agree with the reviewer that, from a synthetic chemistry perspective, M1–M6 would require transition-metal-catalyzed

cross-coupling reactions to construct 2D polymer frameworks, whereas M7 must be synthesized *via* a reversible Aldol-type condensation that enables error correction and the formation of an ideal 2D network. When initially evaluating the conjugation efficiency of TBDT-DPP-based polymer backbones, we also intended to investigate other dynamic covalent linkages commonly used in 2D conjugated COFs—such as imine and imidazole bridges. However, these linkages exhibit significantly poorer π -conjugation and lower chemical stability than the vinylene bridge. In addition, introducing amine or aldehyde functionalities directly onto the DPP core is synthetically extremely challenging, rendering imine (or imine-derivatives) linkages impractical for constructing DPP-based 2D COFs. For these reasons, we did not incorporate such systems into the present work.

Our comparison between M7 and M1–M6 was therefore intended primarily to benchmark the electronic properties of different π -bridging units, rather than to imply equal synthetic feasibility for 2D polymer formation. Notably, M7 exhibits even stronger conjugation than its ring-connected analogues, further motivating our in-depth investigation of the vinylene linkage.

We have revised the text on Page 6 to clarify this rationale. As now stated: “Based on these findings and in view of synthetic feasibility, we propose that developing TBDT/DPP-based (M7 type) 2D PAVs via aldol-type polycondensation would confer efficient 2D conjugation and enhanced charge transport to the 2D polymer frameworks.”

Comment 6: In addition, suitable references for the aldol condensation should be added, as this reaction is the key for constructing the 2D-COF, e.g., *Org. Lett.* 2019, 21, 1973–1978. (<https://pubs.acs.org/doi/10.1021/acs.orglett.9b00019>).

Response 6: We thank you for your suggestion. The reference (*Org. Lett.* 2019, 21, 1973–1978) is cited as Ref. 25.

Ref. 25: D. Feng et al. Synthesis of 2,5-Dibutyl-3,6-dimethyl-1*H*,2*H*,4*H*,5*H*-pyrrolo[3,4-*c*]pyrrole-1,4-dione: A Diketopyrrolopyrrole Scaffold for the Formation of Alkenyldiketopyrrolopyrrole Compounds. *Org. Lett.* **21**, 1973-1978 (2019).

The manuscript reported by Zhao and coworkers describes the synthesis and characterization of DPP-incorporated 2D COF for use in electronic materials. One of the key features of the work is the record-high electron mobility of the materials developed, *i.e.*, 2DPAV-TBDT-DPP-1 and 2DPAV-TBDT-DPP-2. The mobility values are the highest not only for similar 2D-COFs in the field, but also for related 1D conjugated polymers consisting of the DPP units. Thus, provided that the performances are real, the present results are quite appealing and noteworthy. In this regard, evaluating mobility is key, as discussed below.

The mobility was estimated from the equation mentioned in the main text, line 221, where the key parameters are τ_s , c , and m^* . As far as I understand, the former two are extracted from OPTP measurements, and m^* appears to be obtained from theoretical calculations for the band structures. My concern is that the values of m^* obtained from calculations based on the ideal periodic boundary model can easily overestimate m^* , which can be extracted from the curvature in the band dispersion. This may not be realistic for the actual material in the present case. This is due to the low structural perfection of 2DPAV-TBDT-DPP-1 and 2DPAV-TBDT-DPP-2, as indicated by the ^1H and ^{13}C NMR spectra reported in Supplementary Figures 19 and 20. In Supplementary Figure 19, a broad band is observed at around 10 ppm, which can be assigned to the aldehyde moiety of TBDT-1 (and related intermediates). The same is true for the ^{13}C NMR, in which peaks around 180 ppm can also be assigned to the aldehyde moiety. These strongly imply that the structural perfection of 2DPAV-TBDT-DPP-1 and 2DPAV-TBDT-DPP-2 is not that high, and then the m^* s, and accordingly the mobility values, should be carefully reconsidered. Furthermore, the enhanced conductivity resulting from doping with N-DMBI is mentioned in the main text, lines 230-233; however, no quantitative evaluation of electrical conductivity is reported in Supplementary Figure 31. More detailed data should be added.

For these reasons, the authors are advised to evaluate the materials more carefully and should provide clear spectrum data, eliminating any doubt about the characterization of the materials. In this sense, in Supplementary Figures 19-22 (21 and 22 are IR spectra), the spectra of “Model compound 1 (mentioned on page S9 in the Supplementary Information)” should be added to compare with the spectra. Also, the synthetic yield of “Model compound 1” is reported to be 65%, meaning that the efficiency of the basic reaction for constructing the 2D COF is not very high. Considering this and the almost quantitative yields for 2DPAV-TBDT-DPP-1 and 2DPAV-TBDT-DPP-2, the structural characterization of these materials should be conducted with the utmost care.

Another comment to the authors is that I was a little bothered by a feeling of strangeness regarding the description of the “Design of DPP-based donor-acceptor polymer backbone (line 83-100)”. They compared M1-M7 with different p-conjugated monomers, including benzene (M1), pyridine (M2), pyradine (M3), furan (M4), thiophene (M5), selenophene (M6), but for the synthesis of 2D polymer with these aromatic units, different synthetic reactions than the present vinyl (M7) unit, like transition-metal-catalyzed cross coupling reactions, are necessary. From a synthetic chemistry point of view,

the use of the vinyl unit, which is constructed by cross aldol condensation of dimethyl-DPP and aldehyde, followed by dehydration to furnish the vinyl moiety, is only a feasible method to build 2D-COF, as the first aldol condensation is an equilibrium reaction, which allows for to formation of the ideal 2D structure, and then dehydration (non-equilibrium reaction) to fix the 2D-network structure. Thus, the authors are recommended to reconsider this part. In addition, suitable references for the aldol condensation should be added, as this reaction is the key for constructing the 2D-COF, *e.g.*, *Org. Lett.* **2019**, *21*, 1973–1978. (<https://pubs.acs.org/doi/10.1021/acs.orglett.9b00019>).

With revisions and additions concerning the above points, the manuscript could be re-reviewed.